# Non-Transferable Learning: A New Approach for Model Ownership Verification and Applicability Authorization

**Lixu Wang**\*, **Shichao Xu**\*, **Ruiqi Xu, Xiao Wang, Qi Zhu**
Northwestern University
Evanston, IL 60208, USA
{lixuwang2025,shichaoxu2023,jerryxu2023}@u.northwestern.edu,
{wangxiao,qzhu}@northwestern.edu

## Abstract

As Artificial Intelligence as a Service gains popularity, protecting well-trained models as intellectual property is becoming increasingly important. There are two common types of protection methods: *ownership verification* and *usage authorization*. In this paper, we propose **Non-Transferable Learning (NTL)**, a novel approach that captures the exclusive data representation in the learned model and restricts the model generalization ability to certain domains. This approach provides effective solutions to both model verification and authorization. Specifically: 1) For ownership verification, watermarking techniques are commonly used but are often vulnerable to sophisticated watermark removal methods. By comparison, our NTL-based **ownership verification** provides robust resistance to state-of-the-art watermark removal methods, as shown in extensive experiments with 6 removal approaches over the digits, CIFAR10 & STL10, and VisDA datasets. 2) For usage authorization, prior solutions focus on authorizing specific users to access the model, but authorized users can still apply the model to any data without restriction. Our NTL-based authorization approach instead provides data-centric protection, which we call **applicability authorization**, by significantly degrading the performance of the model on unauthorized data. Its effectiveness is also shown through experiments on aforementioned datasets.

## 1 Introduction

Deep Learning (DL) is the backbone of Artificial Intelligence as a Service (AIaaS) (Ribeiro et al., 2015), which is being provided in a wide range of applications including music composition (Briot et al., 2020), autonomous driving (Li et al., 2021a), smart building (Xu et al., 2020a), etc. However, a good model can be expensive to obtain: it often requires dedicated architecture design (He et al., 2016), a large amount of high-quality data (Deng et al., 2009), lengthy training on professional devices (Zoph & Le, 2016), and expert tuning (Zhang et al., 2019). Thus, well-trained DL models are valuable intellectual property (IP) to the model owners and need protection. Generally speaking, there are two aspects in protecting an IP in AIaaS, verifying who owns the model and authorizing how the model can be used. These two aspects led to the development of two types of protection techniques: **ownership verification** and **usage authorization**.

For ownership verification, prior works proposed approaches such as embedding watermarks into network parameters (Song et al., 2017), learning special behaviors for pre-defined triggers (Fan et al., 2019), and extracting fingerprints from the model (Le Merrer et al., 2020). However, they are vulnerable to state-of-art watermark removal approaches that are based on model fine-tuning or retraining (Chen et al., 2019), watermark overwriting and model pruning (Rouhani et al., 2018). For model usage authorization, most prior works were built on encrypting neural network parameters with a secret key (Alam et al., 2020; Chakraborty et al., 2020) and ensuring that models can only be used by users with this key. However, authorized users may use the model on any data without restriction. We believe that for comprehensive IP protection, the goal of usage authorization is not

---

\*These authors contributed equally to this work.

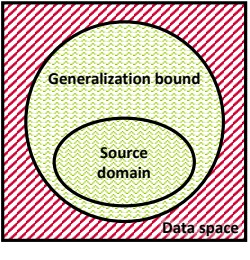 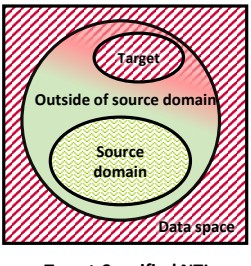 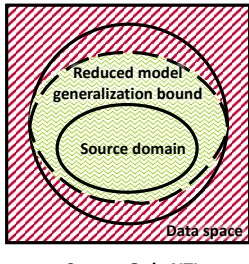

| Supervised Learning | Target-Specified NTL | Source-Only NTL |

Figure 1: **A visualization of the generalization bound trained with different approaches.** The left figure shows Supervised Learning in the source domain, which can derive a wide generalization area. When Target-Specified NTL is applied (middle), the target domain is removed from the generalization area. As for Source-Only NTL (right), the generalization area is significantly reduced.

only who is allowed to use the model, but also *what data can the model be used on*. We thus consider a new data-centric aspect of usage authorization in this work, i.e., authorizing models to certain data for preventing their usage on unauthorized data. We call this **applicability authorization**. Note that applicability authorization goes far beyond IP protection. It can also be viewed as a way to "control" how machine learning models are used in general. One example would be a company (e.g., Meta) trains a recommendation system from adult data and uses applicability authorization to prevent this system from being used by teenagers.

**Our Approach and Contribution.** In this work, we propose **Non-Transferable Learning (NTL)**, a novel approach that can robustly verify the model ownership and authorize the model applicability on certain data. Intuitively, NTL goes against the current research trend of improving the generalization ability of models across various domains, e.g., domain generalization and adaptation (Zhou et al., 2020; Dong et al., 2020). Instead, NTL tries to make the generalization bound of DL models more explicit and narrower, by optimizing the model to learn domain-dependent features and thereby making the model exclusive to certain domains. More specifically, we consider two domains: the source domain where we want the models to perform well, and the auxiliary domain where we aim to degrade the model performance. And if the model trained with NTL is applied to a target domain similar to the auxiliary one, the performance should also be poor. As shown in Figure 1, we have developed two types of NTL approaches: **Target-Specified** NTL and **Source-Only** NTL.

- Target-Specified NTL assumes that the source and target domains are both known. We then treat the target domain as the auxiliary domain and enlarge the distance of representations between the source and auxiliary domains. Target-Specified NTL can be used to verify the model ownership by triggering misclassification. While previous model watermarks can often be easily removed because the model memorization of such watermarks encounters catastrophic forgetting (Kemker et al., 2018) during watermark removal, our NTL-based verification is resistant to state-of-art watermark removal approaches, because the misclassification behavior is dependent on the overall target-private features that have little correlation with the source-private features for the main task.

- In Source-Only NTL, the target domain is unknown and thus our approach relies solely on the source domain, aiming to degrade the performance in all other domains. In this case, NTL generates the auxiliary domain from a novel generative adversarial augmentation framework and then increases the representation distance. Source-Only NTL can provide authorization to certain data rather than particular users or devices, by degrading the model performance on all other data domains other than the source domain. This provides data-centric applicability authorization, with which we can also prevent unauthorized model usage that are caused by the secret key leakage and cannot be addressed by prior model authorization methods.

In addition to proposing the novel concept of NTL and developing its two approaches, we are also able to experimentally validate their effectiveness. We conducted extensive experiments on 5 digit sets, CIFAR10 & STL10 and VisDA. For target-specified cases, we demonstrate how to apply NTL for model ownership verification. Our experiments show that the state-of-art model watermark removal methods are ineffective on NTL-based ownership verification. For source-only NTL, our experiments demonstrate its effectiveness in authorizing model applicability to certain data.

## 2 RELATED WORK

**Domain Generalization & Adaptation (DG & DA).** DG aims to generalize learning models with available source domains to unseen target domains (Blanchard et al., 2011). A number of methods

have been proposed for domain discrepancy minimization (Li et al., 2020), adversarial training (Rahman et al., 2020; Zhao et al., 2020c), invariance representation learning (Zhou et al., 2020; Piratla et al., 2020), etc. Recently, there is significant interest on conducting DG with one source domain only, for which well-crafted data augmentation approaches (Qiao et al., 2020; Zhao et al., 2020b; Li et al., 2021b; Xu et al., 2020b) have been proposed to expand the input space. DA is also related to improving the generalization ability of models across domains (Ahmed et al., 2021), and while DA can access the target data, DG has no access to any target sample (Xu et al., 2021; Dong et al., 2021). Unlike DG or DA, we try to weaken the generalization ability of models by expanding the distance between representations of different domains. Our method works effectively for both the target-specified and the source-only cases with a novel adversarial augmentation framework.

**Intellectual Property (IP) Protection for Deep Learning (DL).** While DL has shown its unparalleled advantages in various applications, there are significant challenges in protecting DL models. For instance, Inference Attack (Shokri et al., 2017; Wang et al., 2019) can steal private information about the target DL model. Model Inversion Attack (He et al., 2019; Salem et al., 2020) is able to recover the input data via an analysis of the model prediction. These two types of attacks directly threaten the privacy of model users, while there are also many active attacks (Suciu et al., 2018; Yao et al., 2019) that lead DL models to produce abnormal behaviors.

In addition, verifying model ownership and authorizing model usage have become important issues with the development of AIaaS. There have been a number of watermarking approaches addressing the verification of model ownership. For instance, Zhang et al. (2018) and Li et al. (2019) train a neural network on the original datasets and the watermarked one assigned with a particular label, which makes the model behave abnormally when it encounters watermarked data. Song et al. (2017) and Uchida et al. (2017) inject a pattern that is similar to regular photograph watermarks (Cheng et al., 2021) into the least significant bits of the model parameters and provide the corresponding decoding methods. Le Merrer et al. (2020) and Zhao et al. (2020a) make use of adversarial examples to extract fingerprints from learned neural networks without accessing network weights. Compared to these approaches, our NTL can achieve model ownership verification by triggering universal misclassification. Moreover, with extensive experiments, we also demonstrate that state-of-art model watermark removal methods, e.g., FTAL and RTAL (Adi et al., 2018), EWC and AU (Chen et al., 2019), watermark overwriting and model pruning (Rouhani et al., 2018) are not effective to NTL-based verification. Model usage authorization is another aspect in protecting model intellectual property. For instance, Alam et al. (2020) encrypt every network parameter with a secret key. Chakraborty et al. (2020) generate a secret key from hardware fingerprints of a particular device, and require that only users who possess this device can load and employ the model. Different from these methods, our NTL focuses on providing data-centric protection via applicability authorization, which retains good model performance on authorized data while degrading model performance for other data domains. To the best of our knowledge, this is the first work that prevents model usage on unauthorized data via model learning.

## 3 METHODOLOGY

In this section, we introduce our NTL approach. Section 3.1 presents the inspiration and the design of the optimization objective of NTL, which is the core for both target-specified and source-only cases. Section 3.2 presents the generative augmentation framework for source-only cases. Our method is based on the concept of generative adversarial networks (GAN), however our goal is not to propose a new GAN but to design an effective augmentation method in the context of NTL. Section 3.3 introduces the application of NTL on ownership verification and applicability authorization.

### 3.1 NON-TRANSFERABLE LEARNING WITH DISTANCE EXPANSION OF REPRESENTATION

We consider a source domain with labeled samples $S = \{(\boldsymbol{x}, \boldsymbol{y}) \| \boldsymbol{x} \sim \mathcal{P}_X^S, \boldsymbol{y} \sim \mathcal{P}_Y^S\}$, where $\mathcal{P}_X$ and $\mathcal{P}_Y$ are the input and label distributions, respectively. In this work, we use image classification as the learning task with $K$ possible classes, in which case $\boldsymbol{x}$ and $\boldsymbol{y}$ are matrix-valued and scalar random variables, respectively. In addition, we consider an auxiliary domain $A = \{(\boldsymbol{x}, \boldsymbol{y}) \| \boldsymbol{x} \sim \mathcal{P}_X^A, \boldsymbol{y} \sim \mathcal{P}_Y^A\}$. The source domain $S$ and the auxiliary domain $A$ will be fed into a deep neural network, and without loss of generality, we split the neural network into two parts, one is a feature extractor $\Phi$ on the bottom, and the other is a classifier $\Omega$ on the top.

**Inspiration from Information Bottleneck.** Our NTL, in particular the design of optimization objective, is inspired by the analysis of *Information Bottleneck* (IB) (Tishby et al., 2000). Let us start by introducing *Shannon Mutual Information* (SMI). In addition to input $\boldsymbol{x}$ and label $\boldsymbol{y}$, we also regard representation $\boldsymbol{z}$ extracted by $\Phi$ as a random variable. The SMI between two random variables, e.g., between $\boldsymbol{z}$ and $\boldsymbol{x}$, is defined as $I(\boldsymbol{z};\boldsymbol{x})=\mathbb{E}_{\boldsymbol{x}\sim\mathcal{P}_X}[D_{\mathrm{KL}}(\mathcal{P}(\boldsymbol{z}|\boldsymbol{x})\|\mathcal{P}(\boldsymbol{z}))]$, where $D_{\mathrm{KL}}(\cdot)$ represents the Kullback-Leible (KL) divergence and $\mathcal{P}(\cdot)$ is the distribution. In IB theory, considering the effectiveness, privacy and generalization, an optimal representation has three properties (Achille & Soatto, 2018): (1) *Sufficiency*: label $\boldsymbol{y}$ sufficiently differentiates representation $\boldsymbol{z}$, i.e., $I(\boldsymbol{z};\boldsymbol{y})=I(\boldsymbol{x};\boldsymbol{y})$; (2) *Minimality*: $\boldsymbol{z}$ needs to represent as little information about input $\boldsymbol{x}$ as possible, i.e., $\min I(\boldsymbol{z};\boldsymbol{x})$; (3) *Invariance*: $\boldsymbol{z}$ is optimal, meaning that it does not overfit to spurious correlations between $\boldsymbol{y}$ and nuisance $\boldsymbol{n}$ embedded in $\boldsymbol{x}$, i.e., $I(\boldsymbol{z};\boldsymbol{n})=0$. IB theory assumes that nuisance $\boldsymbol{n}$ is a factor that affects input $\boldsymbol{x}$, and it works with $\boldsymbol{y}$ together to determine what $\boldsymbol{x}$ looks like to some extent. For instance, in domain generalization, nuisance $\boldsymbol{n}$ can be regarded as a domain index that indicates which domain a certain sample comes from (Du et al., 2020). *In our problem, different from the objective of the IB theory, NTL enforces the models to extract nuisance-dependent representations, which is opposite to the property of invariance.* In other words, we aim to increase $I(\boldsymbol{z};\boldsymbol{n})$, and we have the following proposition for achieving this aim.

**Proposition 1.** *Let $\boldsymbol{n}$ be a nuisance for input $\boldsymbol{x}$. Let $\boldsymbol{z}$ be a representation of $\boldsymbol{x}$, and the label is $\boldsymbol{y}$. For the information flow in the representation learning, we have*

$$I(\boldsymbol{z};\boldsymbol{x}) - I(\boldsymbol{z};\boldsymbol{y}|\boldsymbol{n}) \geq I(\boldsymbol{z};\boldsymbol{n}) \tag{1}$$

The detailed proof for Proposition 1 is included in the Appendix.

**Optimization Objective Design.** Proposition 1 provides guidance for maximizing $I(\boldsymbol{z};\boldsymbol{n})$. First, unlike in the IB theory, we do not minimize $I(\boldsymbol{z};\boldsymbol{x})$ for the minimality property. In addition, we try to minimize $I(\boldsymbol{z};\boldsymbol{y}|\boldsymbol{n})$ through the design of optimization objective that measures the error between the model prediction and the ground truth during the training of neural networks. Specifically, instead of using the typical CrossEntropy loss to measure the error, we apply KL divergence loss to direct the training, and we have the following theorem.

**Theorem 1.** *Let $\hat{\boldsymbol{y}}$ be the predicted label outputted by a representation model when feeding with input $\boldsymbol{x}$, and suppose that $\hat{\boldsymbol{y}}$ is a scalar random variable and $\boldsymbol{x}$ is balanced on the ground truth label $\boldsymbol{y}$. Denote the one-hot forms of $\hat{\boldsymbol{y}}$ and $\boldsymbol{y}$ as $\hat{\mathbf{y}}$ and $\mathbf{y}$, respectively. If the KL divergence loss $D_{KL}(\mathcal{P}(\hat{\mathbf{y}})\|\mathcal{P}(\mathbf{y}))$ increases, the mutual information $I(\boldsymbol{z};\boldsymbol{y})$ will decrease.*

The detailed proof of Theorem 1 is provided in the Appendix. According to this theorem, $I(\boldsymbol{z};\boldsymbol{y}|\boldsymbol{n})$ can be minimized by increasing the KL divergence loss of training data conditioned on different $\boldsymbol{n}$. However, as stated in Section 1, we aim to degrade the model performance in the auxiliary domain while maintaining good model performance in the source domain. Thus, we only minimize $I(\boldsymbol{z};\boldsymbol{y}|\boldsymbol{n})$ by increasing the KL divergence loss of the auxiliary domain data. In order to achieve this goal, we design a loss $L_{ntl}^*$ that shapes like a minus operation between KL divergence losses of the source and auxiliary domain ($L_S$, $L_A$), i.e., $L_S = \mathbb{E}_{\boldsymbol{x}\sim\mathcal{P}_X^S}[D_{\mathrm{KL}}(\mathcal{P}(\Omega(\Phi(\boldsymbol{x})))\|\mathcal{P}(\mathbf{y}))]$ and $L_A = \mathbb{E}_{\boldsymbol{x}\sim\mathcal{P}_X^A}[D_{\mathrm{KL}}(\mathcal{P}(\Omega(\Phi(\boldsymbol{x})))\|\mathcal{P}(\mathbf{y}))]$. Specifically, this loss can be written as follows:

$$L_{ntl}^* = L_S - \min(\beta, \alpha \cdot L_A) \tag{2}$$

Here, $\alpha$ is the scaling factor for $L_A$ ($\alpha = 0.1$ in our experiments), and $\beta$ is an upper bound when $L_A$ gets too large and dominates the overall loss ($\beta = 1.0$ in experiments; please see the Appendix for more details about $\alpha$ and $\beta$). Moreover, if we use $\boldsymbol{n} = 0$ and $\boldsymbol{n} = 1$ to denote the source and auxiliary domain respectively, the optimization of Eq. (2) can guarantee the sufficiency property for the source domain: $I(\boldsymbol{z};\boldsymbol{y}|\boldsymbol{n}=0)=I(\boldsymbol{x};\boldsymbol{y}|\boldsymbol{n}=0)$, and increasing $L_A$ decreases $I(\boldsymbol{z};\boldsymbol{y}|\boldsymbol{n}=1)$.

According to Proposition 1, we can move the upper bound of $I(\boldsymbol{z};\boldsymbol{n})$ to a higher baseline via optimizing Eq. (2). However, such optimization might only make classifier $\Omega$ more sensitive to domain features and have little effect on feature extractor $\Phi$. In this case, representations of different domains captured by $\Phi$ may still be similar, which conflicts with our intention to maximize $I(\boldsymbol{z};\boldsymbol{n})$, and the performance of the target can be easily improved by fine-tuning or adapting $\Omega$ with a small number of labeled target samples. On the other hand, directly calculating $I(\boldsymbol{z};\boldsymbol{n})$ and taking it as a part of the optimization objective are difficult, especially in the optimization of representation learning (Torkkola, 2003). Achille & Soatto (2018) apply binary classifier as the nuisance discriminator, and they can estimate $I(\boldsymbol{z};\boldsymbol{n})$ after the model training via this discriminator. Here, we find another way to increase $I(\boldsymbol{z};\boldsymbol{n})$ indirectly based on the following theorem.

**Theorem 2.** *Let $\boldsymbol{n}$ be a nuisance that is regarded as a domain index. $\boldsymbol{n} = 0$ and $\boldsymbol{n} = 1$ denote that a certain input $\boldsymbol{x}$ comes from two different domains. Suppose that these two domains have the same number of samples $d$, and the samples of each domain are symmetrically distributed around the centroid. Let $\boldsymbol{z}$ be a representation of $\boldsymbol{x}$, and it is drawn from distribution $\mathcal{P}_Z$. An estimator with the characteristic kernel from Reproducing Kernel Hilbert Spaces (RKHSs) – Gaussian Kernel estimator $MMD(\mathcal{P}, \mathcal{Q}; exp)$ is applied on finite samples from distributions $\mathcal{P}_{Z|0}$ and $\mathcal{P}_{Z|1}$ to approximate the Maximum Mean Discrepancy (MMD) between these two distributions. If $MMD(\mathcal{P}_{Z|0}, \mathcal{P}_{Z|1}; exp)$ increases to saturation, the mutual information between $\boldsymbol{z}$ and $\boldsymbol{n}$ will increase.*

$$\text{MMD}(\mathcal{P}_{Z|0}, \mathcal{P}_{Z|1}; exp) = \mathbb{E}_{\boldsymbol{z}, \boldsymbol{z}' \sim \mathcal{P}_{Z|0}}[e^{-\|\boldsymbol{z} - \boldsymbol{z}'\|^2}] - 2\mathbb{E}_{\boldsymbol{z} \sim \mathcal{P}_{Z|0}, \boldsymbol{z}' \sim \mathcal{P}_{Z|1}}[e^{-\|\boldsymbol{z} - \boldsymbol{z}'\|^2}] + \mathbb{E}_{\boldsymbol{z}, \boldsymbol{z}' \sim \mathcal{P}_{Z|1}}[e^{-\|\boldsymbol{z} - \boldsymbol{z}'\|^2}]$$
$$(3)$$

We also employ a nuisance discriminator to observe the change of $I(\boldsymbol{z}; \boldsymbol{n})$ during training. The details of this discriminator design and the proof of Theorem 2 can be found in the Appendix.

**NTL Optimization Objective.** Based on the above analysis, we design our NTL optimization objective to increase $I(\boldsymbol{z}; \boldsymbol{n})$ and extract nuisance-dependent representations. Specifically, we compute the $\text{MMD}(\mathcal{P}, \mathcal{Q}; exp)$ between representations of the source and auxiliary domain data and maximize it. For stability concern, we also set an upper bound to the $\text{MMD}(\mathcal{P}, \mathcal{Q}; exp)$. Then, the overall optimization objective of NTL with distance expansion of representation is shaped as follows:

$$L_{ntl} = L_S - \min(\beta, \alpha \cdot L_A \cdot L_{dis}), \text{ where } L_{dis} = \min(\beta', \alpha' \cdot \text{MMD}(\mathcal{P}_{\boldsymbol{x} \sim \mathcal{P}_X^S}(\Phi(\boldsymbol{x})), \mathcal{P}_{\boldsymbol{x} \sim \mathcal{P}_X^A}(\Phi(\boldsymbol{x})); exp)$$
$$(4)$$

Here, $\alpha'$ and $\beta'$ represent the scaling factor and upper bound of $L_{dis}$ respectively ($\alpha' = 0.1$ and $\beta' = 1.0$ in our experiments; please refer to the Appendix for more details about $\alpha'$ and $\beta'$). $\Phi(\cdot)$ is the feature extractor that outputs the corresponding representations of given inputs.

When the target domain is known and accessible, it will be regarded as the auxiliary domain, and the above NTL with distance expansion of representation can be conducted directly on the source and auxiliary domains. We call such cases *Target-Specified NTL*.

## 3.2 SOURCE DOMAIN AUGMENTATION FOR SOURCE-ONLY NTL

In practice, the target domain might be unknown or unavailable. For such cases, we develop a novel generative augmentation framework to generate an auxiliary domain and then leverage the above NTL process, in what we call *Source-Only NTL*. In the following, we will introduce our augmentation framework, which can generate data samples drawn from the neighborhood distribution of the source domain with different distances and directions to serve as the auxiliary data domain in NTL.

**GAN Design for Source Domain Augmentation.** The overall architecture of our augmentation framework is shaped like a generative adversarial network (GAN) that is made up of a generator $\mathcal{G}$ and a discriminator $\mathcal{D}$. $\mathcal{G}$ takes in normal noise and a label in form of one-hot, and then outputs a data sample. For $\mathcal{D}$, if we feed $\mathcal{D}$ with a sample, it will tell whether this sample is fake or not and predict its label. The adversarial battle happens as $\mathcal{G}$ tries to generate data as real as possible to fool $\mathcal{D}$, while $\mathcal{D}$ distinguishes whether the data being fed to it is real or not to the best of its ability. After sufficient period of such battle, the distributions of the generated data and the ground-truth data are too similar to tell apart (Li et al., 2017). Based on this principle, we utilize $\mathcal{G}$ to approximate the source domain. However, if we follow the standard GAN training, the trained GAN will not generate samples with deterministic labels. ***Therefore, we combine the intuitions of CGAN (Mirza & Osindero, 2014) and infoGAN (Chen et al., 2016) to propose a new training approach for our augmentation framework.*** In our approach, $\mathcal{G}$ uses MSE loss to compare its generated data and the real data. $\mathcal{D}$ consists of three modules: a feature extractor and two classifiers behind the extractor as two branches, where a binary classifier predicts whether the data is real or not, and a multiple classifier outputs the label. Note that these two classifiers both rely on the representations extracted by the feature extractor. For training $\mathcal{D}$, we use MSE loss to evaluate its ability to distinguish real samples from fake ones, and KL divergence to quantify the performance of predicting labels for the real data. Finally, there is an additional training step for enforcing the GAN to generate samples of given labels by optimizing $\mathcal{G}$ and $\mathcal{D}$ simultaneously. The training losses of $L_{\mathcal{G}}$, $L_{\mathcal{D}}$ and $L_{\mathcal{G}, \mathcal{D}}$ are:

$$L_{\mathcal{D}} = \mathbb{E}_{\boldsymbol{x} \sim \mathcal{P}_X^S, \boldsymbol{y} \sim \mathcal{P}_Y^S} \left[ \frac{1}{2}(\|\mathcal{D}_b(\boldsymbol{x}), 1\|^2 + \|\mathcal{D}_b(\mathcal{G}(noise, \boldsymbol{y})), 0\|^2) + D_{\text{KL}}(\mathcal{P}(\mathcal{D}_m(\boldsymbol{x})) \| \mathcal{P}(\boldsymbol{y})) \right]$$
$$L_{\mathcal{G}} = \mathbb{E}_{\boldsymbol{y} \sim \mathcal{P}_Y^S} \left[ \|\mathcal{D}_b(\mathcal{G}(noise, \boldsymbol{y})), 1\|^2 \right], \quad L_{\mathcal{G}, \mathcal{D}} = \mathbb{E}_{\boldsymbol{y}' \sim \mathcal{P}_Y^U}[D_{\text{KL}}(\mathcal{P}(\mathcal{D}_m(\mathcal{G}(noise, \boldsymbol{y}'))) \| \mathcal{P}(\boldsymbol{y}'))]$$
$$(5)$$

---

**Algorithm 1:** Generative Adversarial Data Augmentation for Source-Only NTL

---

**Require:** Source domain data $S = \{(\boldsymbol{x}, \boldsymbol{y}) \| \boldsymbol{x} \sim \mathcal{P}_X^S, \boldsymbol{y} \sim \mathcal{P}_Y^S\}$; Generator $\mathcal{G}$, discriminator $\mathcal{D}$; List of augmentation distance *DIS*, the maximum augmentation direction *DIR*; GAN training epochs $e_{\text{GAN}}$, augmentation training epochs $e_{\text{AUG}}$; Initialize the auxiliary domain data $A = [\,]$;

**Output:** The auxiliary domain data $A = \{(\boldsymbol{x}, \boldsymbol{y}) \| \boldsymbol{x} \sim \mathcal{P}_X^A, \boldsymbol{y} \sim \mathcal{P}_Y^A\}$;

1   **for** $i = 1$ *to* $e_{GAN}$ **do**
2     use $(noise, \boldsymbol{y} \sim \mathcal{P}_Y^S)$ to optimize $\mathcal{G}$ with $L_{\mathcal{G}}$, use $S$ and $\mathcal{G}(noise, \boldsymbol{y} \sim \mathcal{P}_Y^S)$ to optimize $\mathcal{D}$ with $L_{\mathcal{D}}$;
3     use $(noise, \boldsymbol{y} \sim \mathcal{P}_Y^U)$ to optimize $\mathcal{G}, \mathcal{D}$ with $L_{\mathcal{G},\mathcal{D}}$;
4   **for** $dis$ *in* $DIS$ **do**
5     **for** $dir$ *to* $DIR$ **do**
6       **for** $l$ *in* $\mathcal{G}$ **do**
7         $interval = $ d($l$) / *DIR*; // *function d() acquires the dimension of inputs*;
8         freeze $\mathcal{D}$ and $l[0 : dir \times interval]$; // *freeze $\mathcal{D}$, and the first dir parts of l-th layer in $\mathcal{G}$*;
9       **for** $i = 1$ *to* $e_{AUG}$ **do**
10         use $(noise, \boldsymbol{y} \sim \mathcal{P}_Y^S)$ and $S$ to optimize $\mathcal{G}$ with $L_{aug}$;
11       $A \cup \mathcal{G}(noise, \boldsymbol{y} \sim \mathcal{P}_Y^U)$; // *use $\mathcal{G}$ to generate augmentation data*;

---

Here, we use subscripts $b$ and $m$ to denote outputs from the binary classifier and the multiple classifier of $\mathcal{D}$, respectively. The $noise$ of Eq. (5) is drawn from Gaussian Noise $\mathcal{P}_g = \mathcal{N}(0, 1)$, while $\boldsymbol{y}'$ is drawn from the uniform distribution $\mathcal{P}_Y^U$ with $K$ equally likely possibilities. And $\mathbf{y}$ and $\mathbf{y}'$ are the one-hot form vectors of $\boldsymbol{y}$ and $\boldsymbol{y}'$, respectively.

**Augmentation with Different Distances.** To generate the data of different distances to the source domain, we apply a Gaussian estimator to measure the MMD between distributions of the source and the generated data from $\mathcal{G}$. However, providing that the MMD distance is optimized to increase with no restriction, the outcome will lose the semantic information, i.e., the essential feature for the main task. In order to preserve such semantic information, we use the CrossEntropy loss to add a restriction to the optimization objective. With this restriction, we set multiple upper bounds – *DIS* for generating data with different distances (we use *DIS* to denote a list of multiple $dis$-s with various lengths). The specific objective is as follows:

$$
\begin{aligned}
L_{aug} = & -\min\left\{dis, \text{MMD}(\mathcal{P}_{\boldsymbol{x} \sim \mathcal{P}_X^S}(\mathcal{D}_z(\boldsymbol{x})), \mathcal{P}_{\boldsymbol{y} \sim \mathcal{P}_Y^S}(\mathcal{D}_z(\mathcal{G}(noise, \boldsymbol{y}))); exp)\right\} \\
& + \mathbb{E}_{\boldsymbol{y} \sim \mathcal{P}_Y^S} D_{\text{CE}}(\mathcal{D}_m(\mathcal{G}(noise, \boldsymbol{y})), \boldsymbol{y})
\end{aligned}
\tag{6}
$$

Here, subscript $z$ denotes outputs from the feature extractor. For every $dis$, we freeze $\mathcal{D}$ and use Eq. (6) to optimize $\mathcal{G}$. After the optimization, we can generate augmentation data via feeding $\mathcal{G}$ with normal noise and labels drawn from $\mathcal{P}_Y^U$.

**Augmentation with Different Directions.** We also investigate how to generate data in different directions. The optimization of Gaussian MMD follows the direction of gradient, which is the fastest way to approach the objective. In such case, all augmented domains of different distances might follow the same direction, i.e., the direction of gradient. Therefore, in order to augment neighborhood domains with different directions, we need to introduce more restrictions to the optimization process. Specifically, for intermediate representations of $\mathcal{G}$, we view each filter (neuron) as corresponding to a feature dimension of the representation. At the beginning of directional augmentation, we make multiple copies of the trained GAN in the last step ($\mathcal{G}$ and $\mathcal{D}$), and we pick one GAN for each direction. If we want to augment the source in *DIR* directions, we will divide the overall network of $\mathcal{G}$ into *DIR* parts equally. For the augmentation of the first direction, the first part of $\mathcal{G}$ will be frozen and not updated during optimization. The second direction will be augmented by freezing the first two parts of the network and conducting the optimization. The third corresponds to the first three parts, and so on. Given a certain $dis$, we can optimize Eq. (6) to augment the source domain to DIR directions by freezing $\mathcal{G}$ gradually. The detained flow is shown in Algorithm 1.

### 3.3 APPLICATION OF NTL FOR MODEL INTELLECTUAL PROPERTY PROTECTION

**Ownership Verification.** The proposed NTL can easily verify the ownership of a learning model by triggering misclassification. To achieve this, we can attach a certain trigger patch which shapes like a shallow mask for images on the source domain data as the auxiliary domain data, and then conduct NTL on these two domains. After that, the training model will perform poorly on the data with the patch but have good performance on the data without the patch. Such model behavior

Table 1: *The performance difference of digit datasets between Supervised Learning and Target-Specified NTL.* The left of '⇒' is the precision (%) in the target when the model is trained on the source dataset with Supervised Learning. The right of '⇒' is the precision of the model trained with Target-Specified NTL. The last two columns present the average relative performance drop in the source and target respectively. It shows that NTL can degrade the performance in the target domains while retaining good performance in the source.

| Source/Target | MT | US | SN | MM | SD | Avg Drop (Source) | Avg Drop (Target) |
|---|---|---|---|---|---|---|---|
| MT | 98.9⇒97.9 | 86.4⇒14.5 | 33.3⇒13.1 | 57.4⇒ 9.8 | 35.7⇒ 8.9 | 1.01% | 78.24% |
| US | 84.7⇒ 8.6 | 99.8⇒98.8 | 26.8⇒ 8.6 | 31.5⇒ 9.8 | 37.5⇒ 8.8 | 1.00% | 80.17% |
| SN | 52.0⇒ 9.5 | 69.0⇒14.2 | 89.5⇒88.4 | 34.7⇒10.2 | 55.1⇒11.6 | 1.23% | 78.42% |
| MM | 97.0⇒11.7 | 80.0⇒13.7 | 47.8⇒19.1 | 91.3⇒89.2 | 45.5⇒10.2 | 2.30% | 79.76% |
| SD | 60.4⇒10.7 | 74.6⇒ 6.8 | 37.8⇒ 8.3 | 35.0⇒12.5 | 97.2⇒96.1 | 1.13% | 81.57% |

difference can be utilized to verify the ownership, which is similar to what regular backdoor-based model watermarking approaches do. In contrast, models trained with other methods often present nearly the same performance on the data with or without the patch, as the patch is shallow and light.

**Applicability Authorization.** When applying NTL to authorize the model applicability, we aim to restrict the model generalization ability to only the authorized domain, where all the data is attached with a dedicated patch, and we need to make sure that the patch design will not impact the semantic information (note that the unforgeability and uniqueness of the patch are not the main consideration of this work, and we will explore it in the future). For simplicity, we use a shallow mask that is similar to that of the aforementioned ownership verification as the authorized patch. We first use our generative adversarial augmentation framework to generate neighborhood data of the original source domain. Then, we regard the source data with the patch attached as the source domain in NTL, and use the union of the original source data, the generated neighborhood data with and without the patch as the auxiliary domain. After the NTL training on these two domains, the learning model performs well only on the source domain data with the authorized patch attached, and exhibits low performance in other domains. In this way, we achieve the model applicability authorization.

## 4 EXPERIMENTAL RESULTS

Our code is implemented in PyTorch (and provided in the Supplementary Materials). All experiments are conducted on a server running Ubuntu 18.04 LTS, equipped with NVIDIA TITAN RTX GPU. The datasets and experiment settings used are introduced below.

**Digits.** MNIST (Deng, 2012) (*MT*) is the most popular digit dataset. USPS (Hull, 1994) (*US*) consists of digits that are scanned from the envelopes by the U.S. Postal Service. SVHN (Netzer et al., 2011) (*SN*) contains house number data selected from Google Street View images. MNIST-M (Ganin et al., 2016) (*MM*) is made by combining MNIST with different backgrounds. Finally, SYN-D (Roy et al., 2018) (*SD*) is a synthetic dataset, which is generated by combining noisy and complex backgrounds. **CIFAR10 & STL10:** Both CIFAR10 and STL10 (Coates et al., 2011) are ten-class classification datasets. In order to make these two sets applicable to our problem, we follow the procedure in French et al. (2017). **VisDA:** This dataset (Peng et al., 2017) contains a training set (*VisDA-T*) and a validation set (*VisDA-V*) of 12 object categories. For classifying these datasets, we apply VGG-11 (Simonyan & Zisserman, 2014) for Digits Recognition, VGG-13 (Simonyan & Zisserman, 2014) for CIFAR10 & STL10, and ResNet-50 (He et al., 2016) and VGG-19 for VisDA. All networks are initialized as the pre-trained version of ImageNet (Deng et al., 2009). We use 3 seeds (2021, 2022, 2023) to conduct all experiments three times and present the average performance. Network architectures, parameters and error bars can be found in the Appendix.

### 4.1 TARGET-SPECIFIED NTL

**Effectiveness of NTL in Reducing Target Domain Performance.** For digits sets and CIFAR10 & STL10, we pick all possible domain pairs to carry out experiments. As for VisDA, we regard the training set as the source domain and the validation set as the target. We include the results of standard supervised learning with KL divergence in the source domain and report the performance difference provided by using NTL. Table 1 and Figure 3 show results of digits sets, CIFAR10 & STL10, and VisDA. For NTL, we observe that the target performance of all pairs is degraded to nearly 10% with little accuracy reduction in the source domain. The largest performance degradation of the target domain, from 97.0% to 11.7%, occurs when the source is MM and the target is MT. Comparing NTL with supervised learning, the average relative performance degradation for the tar-

Table 2: *The results of Target-Specified NTL on ownership verification and the performance after applying different watermark removal methods.* Compared to Supervised Learning, models trained with Target-Specified NTL behave differently depending on whether the data fed to it contains a trigger patch. Applying 6 state-of-art watermark removal approaches on models of NTL does not impact the effectiveness of ownership verification.

| Source without Patch | Training Methods | | Watermark Removal Approaches | | | | | |
|---|---|---|---|---|---|---|---|---|
| | Supervised Learning | NTL | FTAL | RTAL | EWC | AU | Overwriting | Pruning |
| | [Test with/without Patch (%)] | | [Test with/without Patch (%)] | | | | | |
| MT | 99.5/99.5 | 10.2/98.9 | 9.9/99.3 | 10.4/98.6 | 10.9/99.2 | 10.8/98.9 | 11.0/99.0 | 10.0/87.3 |
| US | 99.1/99.2 | 14.3/99.2 | 14.3/99.0 | 9.1/98.6 | 14.4/98.8 | 14.2/99.1 | 13.9/99.0 | 12.7/88.0 |
| SN | 87.2/89.3 | 10.1/89.0 | 9.9/89.2 | 10.0/88.7 | 10.1/89.1 | 10.0/89.0 | 9.9/88.8 | 9.7/56.7 |
| MM | 84.8/91.8 | 12.1/90.5 | 14.0/91.0 | 15.1/89.6 | 12.7/91.1 | 12.6/91.0 | 12.5/90.9 | 10.9/68.2 |
| SD | 87.4/96.8 | 11.6/96.5 | 12.6/96.7 | 13.3/95.4 | 12.4/96.9 | 12.8/96.5 | 12.2/96.5 | 11.0/72.4 |
| CIFAR10 | 81.7/89.2 | 14.8/88.9 | 13.5/89.0 | 14.9/88.8 | 15.0/89.4 | 14.8/89.5 | 14.3/88.8 | 12.9/79.2 |
| STL10 | 85.0/87.0 | 14.9/86.2 | 12.4/86.8 | 12.7/86.9 | 13.0/87.5 | 13.9/87.4 | 13.7/87.2 | 11.7/76.3 |
| VisDA-T | 91.7/92.8 | 15.4/92.4 | 15.6/92.7 | 15.7/92.5 | 16.2/92.7 | 16.3/92.6 | 14.7/92.0 | 14.5/82.1 |

STL10 without Patch

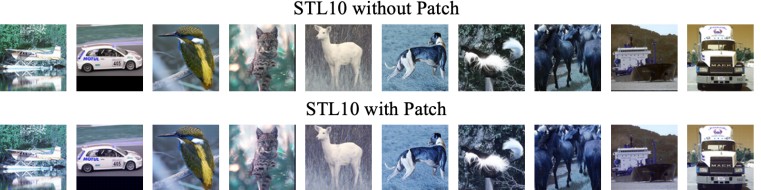

STL10 with Patch

Figure 2: The data of STL10 attached without/with the patch.

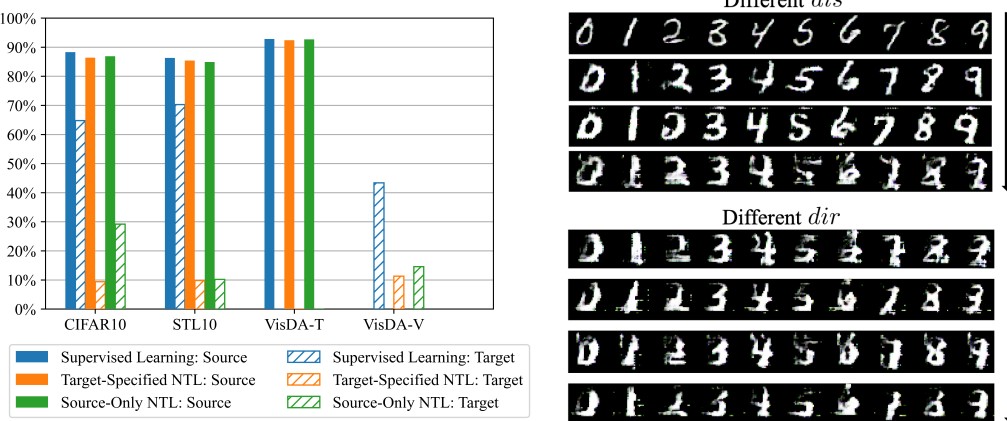

Figure 3: Performance of CIFAR10, STL10, VisDA as source or target domain for Supervised Learning, Target-Specified NTL and Source-Only NTL.

Figure 4: The augmentation data of MNIST generated by the generative adversarial augmentation framework.

get domain of all cases is approximately $80\%$. These results demonstrate that Target-Specified NTL can effectively degrade the performance in the target without sacrificing the source performance.

**NTL for Ownership Verification.** We use a simple pixel-level mask as the trigger patch, which is shown in Figure 2 (please refer to the Appendix for more details). We use 6 state-of-art model watermark removal approaches to test the robustness of NTL-based verification: FTAL (Adi et al., 2018), RTAL (Adi et al., 2018), EWC (Chen et al., 2019), AU (Chen et al., 2019), watermark overwriting and model pruning (Rouhani et al., 2018). The settings of these methods are included in the Appendix. The results are shown in Table 2. We can see that models trained with NTL behave differently on the data with and without the patch, whereas supervised learning performs nearly the same. Furthermore, all these 6 watermark removal methods fail to improve the performance on the patched data, which indicates that NTL-based ownership verification is effective and robust to state-of-the-art watermark removal methods.

## 4.2 SOURCE-ONLY NTL

**Effectiveness of NTL in Reducing Non-source Domain Performance.** For all three dataset cases, we select one domain as the source and then conduct our generative adversarial augmentation to

Table 3: *The performance difference of digit datasets between Supervised Learning and Source-Only NTL. Non-S means non-source domain. The left of '⇒' shows the precision (%) of the model trained on the source dataset with Supervised Learning. The right is the model precision using Source-Only NTL. The last two columns present the average relative precision drop on the source and non-source domain. It shows that Source-Only NTL can effectively degrade the performance in non-source domains without significant sacrifice of source performance.*

| Source/Non-S | MT | US | SN | MM | SD | Avg Drop (Source) | Avg Drop (Non-S) |
|---|---|---|---|---|---|---|---|
| MT | 98.9⇒98.9 | 86.4⇒13.8 | 33.3⇒20.8 | 57.4⇒13.4 | 35.7⇒11.0 | 0.00% | 72.27% |
| US | 84.7⇒ 6.7 | 99.8⇒98.9 | 26.8⇒ 6.0 | 31.5⇒10.1 | 37.5⇒ 8.6 | 0.90% | 82.60% |
| SN | 52.0⇒12.3 | 69.0⇒ 8.9 | 89.5⇒88.0 | 34.7⇒11.3 | 55.1⇒12.7 | 1.68% | 78.56% |
| MM | 97.0⇒14.7 | 80.0⇒ 7.8 | 47.8⇒ 7.8 | 91.3⇒89.1 | 45.5⇒20.1 | 2.41% | 81.35% |
| SD | 60.4⇒39.2 | 74.6⇒ 9.5 | 37.8⇒11.4 | 35.0⇒20.7 | 97.2⇒96.9 | 0.31% | 61.12% |

Table 4: *The performance of authorizing applicability of models trained with Source-Only NTL on digits. NTL-based model authorization can enable the model to perform well only on the authorized domain – the source data attached with the authorized patch.*

| Source with Patch | Test with Patch(%) | | | | | Test without Patch(%) | | | | | Authorized Domain | Other Domains |
|---|---|---|---|---|---|---|---|---|---|---|---|---|
| | MT | US | SN | MM | SD | MT | US | SN | MM | SD | | |
| MT | 98.5 | 14.2 | 17.7 | 15.1 | 7.7 | 11.6 | 14.3 | 16.2 | 10.7 | 9.3 | 98.5 | 13.0 |
| US | 8.2 | 98.9 | 8.7 | 9.5 | 10.4 | 10.3 | 6.6 | 8.6 | 9.2 | 10.3 | 98.9 | 9.1 |
| SN | 10.1 | 9.7 | 88.9 | 12.9 | 11.8 | 9.7 | 8.3 | 9.2 | 12.3 | 11.5 | 88.9 | 10.6 |
| MM | 42.7 | 8.2 | 16.1 | 90.6 | 32.2 | 9.5 | 8.3 | 7.1 | 25.6 | 22.8 | 90.6 | 19.2 |
| SD | 9.7 | 6.7 | 15.7 | 19.2 | 95.8 | 10.2 | 6.7 | 9.1 | 11.5 | 33.7 | 95.8 | 13.6 |

generate the auxiliary domain. We set a series of discrete $dis$-s from 0.1 to 0.5 with a step of 0.1, and for each $dis$, we generate augmentation data of 4 directions (DIR=4). Table 3 and Figure 3 present the results of Source-Only NTL and its comparison with the supervised learning. Figure 4 is the augmentation data for MNIST (other datasets are included in the Appendix). From the results, we can clearly see that models trained with NTL perform worse on all non-source domains compared with the supervised learning, and MM-MT has the largest degradation from 97.0% to 14.7%.

**NTL for Applicability Authorization.** Follow the implementation steps outlined in Section 3.3, we carry out experiments on all 3 dataset cases. The experiment results of digits are presented in Table 4 (the results of CIFAR10 & STL10 and VisDA are in the Appendix). From the table, we can see that the model performs very well in the authorized domain while having bad performance in all other domains (with or without the authorized patch). The highest classification accuracy of unauthorized domains is barely 42.7%, which will discourage users from employing this model. This shows the effectiveness of NTL in applicability authorization.

## 5 CONCLUSION AND FUTURE WORK

In this paper, we propose Non-Transferable Learning (NTL), a novel training approach that can restrict the generalization ability of deep learning models to a specific data domain while degrading the performance in other domains. With the help of a generative adversarial augmentation framework, NTL is effective both in the presence and absence of target domains. Extensive experiments on 5 digit recognition datasets, CIFAR10 & STL10 and VisDA demonstrate that the ownership of models trained with NTL can be easily verified, and the verification is resistant to state-of-art watermark removal approaches. Moreover, with the training of NTL, model owners can authorize the model applicability to a certain data domain without worrying about unauthorized usage in other domains.

In future work, it would be interesting to extend NTL to other tasks beyond image classification, e.g., semantic segmentation, object detection/tracking, and natural language processing (NLP) tasks. For tasks where the input data is not images, generating augmentation data would require different methods and could be challenging. Another possible future direction is Multi-Task Learning, where we could explore whether it is possible to restrict the model generalization ability to certain tasks, for instance, we think it might be useful in some cases to restrict the language model to certain tasks. Moreover, yet another interesting direction could be to combine cryptography with NTL-based verification and authorization.

## ACKNOWLEDGEMENT

We gratefully acknowledge the support by National Science Foundation grants 1834701, 1724341, 2038853, 2016240, Office of Naval Research grant N00014-19-1-2496, and research awards from Facebook, Google, PlatON Network, and General Motors.

## ETHICS STATEMENT

In this paper, our studies are not related to human subjects, practices to data set releases, discrimination/bias/fairness concerns, and also do not have legal compliance or research integrity issues. Non-Transferable Learning is proposed to address the shortcomings of current learning model in intellectual property protection. However, if the model trainer themselves is malicious, they may utilize NTL for harmful purposes. For example, a malicious trainer could use NTL to implant backdoor triggers in their model and release the model to the public. In addition, recently, there are domain adaptation (DA) works on adapting the domain-shared knowledge within the source model to the target one without access to the source data (Liang et al., 2020; Ahmed et al., 2021; Kundu et al., 2020; Wang et al., 2021). However, if the source model is trained with NTL, we believe that these DA approaches will be ineffective. In other words, our NTL can be regarded as a type of attack to those source-free DA works.

## REPRODUCIBILITY STATEMENT

The implementation code can be found in https://github.com/conditionWang/NTL. All datasets and code platform (PyTorch) we use are public. In addition, we also provide detailed experiment parameters and random seeds in the Appendix.

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

# SUMMARY OF THE APPENDIX

This appendix contains additional details for the ICLR 2022 article *"Non-Transferable Learning: A New Approach for Model Ownership Verification and Applicability Authorization"*, including mathematical proofs, experimental details and additional results. The appendix is organized as follows:

- Section A introduces the theoretical proofs for Proposition 1, Theorem 1 and Theorem 2.

- Section B provides additional implementation settings, including the network architectures (Section B.1) and hyper parameters (Section B.2).

- Section C provides additional experimental results, including the augmentation data of other datasets (Section C.1), the model authorization results on CIFAR10 & STL10 and VisDA (Section C.2), the experiments of VisDA on VGG-19 (Section C.3), and the error bars of main experiment results (Section C.4). Section C.5 provides the experiment result of different kernel widths.

- In Section D, we discuss possible attacks that can be constructed based on our proposed method. Note that *NTL* used in this appendix is the abbreviation of *Non-Transferable Learning*.

## A  THEORY PROOFS

### A.1  PROOF

***Proposition 1.*** *Let $\boldsymbol{n}$ be a nuisance for input $\boldsymbol{x}$. Let $\boldsymbol{z}$ be a representation of $\boldsymbol{x}$, and the label is $\boldsymbol{y}$. For the information flow in the representation learning, we have*

$$I(\boldsymbol{z};\boldsymbol{x}) - I(\boldsymbol{z};\boldsymbol{y}|\boldsymbol{n}) \geq I(\boldsymbol{z};\boldsymbol{n}) \tag{7}$$

**Proof:** According to Proposition 3.1 in (Achille & Soatto, 2018), there is a Markov Chain: $(\boldsymbol{y},\boldsymbol{n}) \to \boldsymbol{x} \to \boldsymbol{z}$. This chain describes the information flow starting from the ground truth knowledge (label $\boldsymbol{y}$ and nuisance $\boldsymbol{n}$) of input $\boldsymbol{x}$ to extracted representation $\boldsymbol{z}$. In this case, the information flows from $(\boldsymbol{y},\boldsymbol{n})$ to $\boldsymbol{x}$ then to $\boldsymbol{z}$. The Data Processing Inequality (DPI) for a Markov Chain can ensure the relation that $I(\boldsymbol{z};\boldsymbol{x}) \geq I(\boldsymbol{z};\boldsymbol{y},\boldsymbol{n})$. And with the chain rule, we have $I(\boldsymbol{z};\boldsymbol{y},\boldsymbol{n}) = I(\boldsymbol{z};\boldsymbol{n}) + I(\boldsymbol{z};\boldsymbol{y}|\boldsymbol{n})$. Thus, we can obtain $I(\boldsymbol{z};\boldsymbol{x}) \geq I(\boldsymbol{z};\boldsymbol{n}) + I(\boldsymbol{z};\boldsymbol{y}|\boldsymbol{n})$. ∎

***Theorem 1.*** *Let $\hat{\boldsymbol{y}}$ be the predicted label outputted by a representation model when feeding with input $\boldsymbol{x}$, and suppose that $\hat{\boldsymbol{y}}$ is a scalar random variable and $\boldsymbol{x}$ is balanced on the ground truth label $\boldsymbol{y}$. Denote the one-hot forms of $\hat{\boldsymbol{y}}$ and $\boldsymbol{y}$ as $\hat{\mathbf{y}}$ and $\mathbf{y}$, respectively. If the KL divergence loss $D_{KL}(\mathcal{P}(\hat{\mathbf{y}})\|\mathcal{P}(\mathbf{y}))$ increases, the mutual information $I(\boldsymbol{z};\boldsymbol{y})$ will decrease.*

**Proof.** Suppose that the information flow in classifier $\Omega$ of a representation model follow a Markov chain $\boldsymbol{z} \to \hat{\boldsymbol{y}} \to \boldsymbol{y}$, in this case, let's apply Data Processing Inequality, we have

$$I(\boldsymbol{z};\boldsymbol{y}) \leq I(\hat{\boldsymbol{y}};\boldsymbol{y}) = \mathbb{E}_{\boldsymbol{y}\sim\mathcal{P}_Y}[D_{\mathrm{KL}}(\mathcal{P}(\hat{\boldsymbol{y}}|\boldsymbol{y})\|\mathcal{P}(\hat{\boldsymbol{y}}))] \tag{8}$$

Because the input data of this representation model is balanced across classes, we suppose that both $\boldsymbol{y}$ and $\hat{\boldsymbol{y}}$ are drawn from the uniform distribution $\mathcal{P}_Y^U$ with $K$ equally likely possibilities. Moreover, though the distribution of $\hat{\boldsymbol{y}}$ might change a little during training, we can assume it won't become very biased since the balance of input data. For both $\hat{\mathbf{y}}$ and $\mathbf{y}$, they are vectors with $K$ dimensions. In PyTorch implementation, the computation of KL divergence loss regards there is a scalar random variable corresponding to each vector and every dimension within the vector as an observation of this variable. In this case, the loss between $\hat{\mathbf{y}}$ and $\mathbf{y}$ forms like $D_{\mathrm{KL}}(\mathcal{P}(\hat{\mathbf{y}})\|\mathcal{P}(\mathbf{y})) = \sum_{i=1}^{K} \hat{\mathbf{y}}_i \cdot \log \frac{\hat{\mathbf{y}}_i}{\mathbf{y}_i}$. It is easy to obtain that $D_{\mathrm{KL}}(\mathcal{P}(\hat{\mathbf{y}})\|\mathcal{P}(\mathbf{y}))$ is non-negative, and if and only if $\hat{\mathbf{y}} = \mathbf{y}$, the KL divergence loss hits the minimum value $D_{\mathrm{KL}}(\mathcal{P}(\hat{\mathbf{y}})\|\mathcal{P}(\mathbf{y})) = 0$. While $\hat{\boldsymbol{y}}$ and $\boldsymbol{y}$ are scalar random variables that equal to the dimension index with the maximum value of $\hat{\mathbf{y}}$ and $\mathbf{y}$, respectively. Therefore, it's easy to conclude that the probability of $\hat{\boldsymbol{y}} = \boldsymbol{y}$ will decrease with the increase of $D_{\mathrm{KL}}(\mathcal{P}(\hat{\mathbf{y}})\|\mathcal{P}(\mathbf{y}))$, i.e., $D_{\mathrm{KL}}(\mathcal{P}(\hat{\mathbf{y}})\|\mathcal{P}(\mathbf{y})) \uparrow \Rightarrow \mathcal{P}(\hat{\boldsymbol{y}},\boldsymbol{y}) \downarrow$. For Eq. (8), we can derive it deeper

$$I(\hat{\boldsymbol{y}};\boldsymbol{y}) = \mathbb{E}_{\boldsymbol{y}\sim\mathcal{P}_Y}[D_{\mathrm{KL}}(\mathcal{P}(\hat{\boldsymbol{y}}|\boldsymbol{y})\|\mathcal{P}(\hat{\boldsymbol{y}}))] = \sum_{\boldsymbol{y}} \mathcal{P}(\boldsymbol{y}) \cdot \sum_{\hat{\boldsymbol{y}}} \frac{\mathcal{P}(\hat{\boldsymbol{y}},\boldsymbol{y})}{\mathcal{P}(\hat{\boldsymbol{y}})} \log \frac{\mathcal{P}(\hat{\boldsymbol{y}},\boldsymbol{y})}{\mathcal{P}(\hat{\boldsymbol{y}}) \cdot \mathcal{P}(\boldsymbol{y})} \tag{9}$$

Here, both $\mathcal{P}(\hat{\boldsymbol{y}})$ and $\mathcal{P}(\boldsymbol{y})$ are uniform distributions $\mathcal{P}_Y^U$, and we have assumed $\mathcal{P}(\hat{\boldsymbol{y}})$ won't become very biased at the beginning of this proof. As a result, we regard $\mathcal{P}(\hat{\boldsymbol{y}})$ and $\mathcal{P}(\boldsymbol{y})$ nearly unchanged after training. In addition, $\mathcal{P}(\hat{\boldsymbol{y}},\boldsymbol{y})$ decreases with the increase of $D_{\mathrm{KL}}(\mathcal{P}(\hat{\mathbf{y}})\|\mathcal{P}(\mathbf{y}))$. Furthermore, we can easily calculate that $\frac{\partial I(\hat{\boldsymbol{y}};\boldsymbol{y})}{\partial \mathcal{P}(\hat{\boldsymbol{y}},\boldsymbol{y})} < 0$. In this case, $I(\hat{\boldsymbol{y}};\boldsymbol{y})$ decreases with the increase of $D_{\mathrm{KL}}(\mathcal{P}(\hat{\mathbf{y}})\|\mathcal{P}(\mathbf{y}))$, and the same as $I(\boldsymbol{z};\boldsymbol{y})$ since $I(\hat{\boldsymbol{y}};\boldsymbol{y})$ is the upper bound. ∎

**Theorem 2.** *Let $\boldsymbol{n}$ be a nuisance that is regarded as a domain index. $\boldsymbol{n} = 0$ and $\boldsymbol{n} = 1$ denote that a certain input $\boldsymbol{x}$ comes from two different domains. Suppose that these two domains have the same number of samples $d$, and the samples of each domain are symmetrically distributed around the centroid. Let $\boldsymbol{z}$ be a representation of $\boldsymbol{x}$, and it is drawn from distribution $\mathcal{P}_Z$. An estimator with the characteristic kernel from Reproducing Kernel Hilbert Spaces (RKHSs) – Gaussian Kernel estimator $MMD(\mathcal{P}, \mathcal{Q}; exp)$ is applied on finite samples from distributions $\mathcal{P}_{Z|0}$ and $\mathcal{P}_{Z|1}$ to approximate the Maximum Mean Discrepancy (MMD) between these two distributions. If $MMD(\mathcal{P}_{Z|0}, \mathcal{P}_{Z|1}; exp)$ increases to saturation, the mutual information between $\boldsymbol{z}$ and $\boldsymbol{n}$ will increase.*

$$\text{MMD}(\mathcal{P}_{Z|0}, \mathcal{P}_{Z|1}; exp) = \mathbb{E}_{\boldsymbol{z}, \boldsymbol{z}' \sim \mathcal{P}_{Z|0}}[e^{-\|\boldsymbol{z}-\boldsymbol{z}'\|^2}] - 2\mathbb{E}_{\boldsymbol{z} \sim \mathcal{P}_{Z|0}, \boldsymbol{z}' \sim \mathcal{P}_{Z|1}}[e^{-\|\boldsymbol{z}-\boldsymbol{z}'\|^2}] + \mathbb{E}_{\boldsymbol{z}, \boldsymbol{z}' \sim \mathcal{P}_{Z|1}}[e^{-\|\boldsymbol{z}-\boldsymbol{z}'\|^2}]$$
(10)

**Proof.** According to the definition of *Shannon Mutual Information*, we have

$$I(\boldsymbol{z}; \boldsymbol{n}) = \mathbb{E}_{\boldsymbol{n} \sim \mathcal{P}(\boldsymbol{n})} D_{\text{KL}}(\mathcal{P}(\boldsymbol{z}|\boldsymbol{n}) \| \mathcal{P}(\boldsymbol{z})) = \mathbb{E}_{\boldsymbol{n} \sim \mathcal{P}(\boldsymbol{n})} \mathbb{E}_{\boldsymbol{z} \sim \mathcal{P}(\boldsymbol{z}|\boldsymbol{n})} \log \frac{\mathcal{P}(\boldsymbol{z}|\boldsymbol{n})}{\mathcal{P}(\boldsymbol{z})} \quad (11)$$

And because two domains have the same number of samples, we can have $\boldsymbol{n}$ conforms $\mathcal{P}(\boldsymbol{n}) \sim \{\mathcal{P}(0) = 0.5, \mathcal{P}(1) = 0.5\}$, Eq. (11) can re-written as

$$I(\boldsymbol{z}; \boldsymbol{n}) = 0.5 \mathbb{E}_{\boldsymbol{z} \sim \mathcal{P}(\boldsymbol{z}|\boldsymbol{n}=0)} \log \frac{\mathcal{P}(\boldsymbol{z}|\boldsymbol{n}=0)}{\mathcal{P}(\boldsymbol{z})} + 0.5 \mathbb{E}_{\boldsymbol{z} \sim \mathcal{P}(\boldsymbol{z}|\boldsymbol{n}=1)} \log \frac{\mathcal{P}(\boldsymbol{z}|\boldsymbol{n}=1)}{\mathcal{P}(\boldsymbol{z})} \quad (12)$$

Next, we denote the probability density function (PDF) of $\mathcal{P}_{Z|0}$ and $\mathcal{P}_{Z|1}$ as $p(\boldsymbol{z})$ and $q(\boldsymbol{z})$, respectively. Moreover, according to the law of total probability, the PDF of distribution $\mathcal{P}_Z$ is $\text{PDF}(\mathcal{P}_Z) = 0.5p(\boldsymbol{z}) + 0.5q(\boldsymbol{z})$, then we have

$$I(\boldsymbol{z}; \boldsymbol{n}) = 0.5 \int_{-\infty}^{+\infty} p(\boldsymbol{z}) \cdot \log \frac{2p(\boldsymbol{z})}{p(\boldsymbol{z}) + q(\boldsymbol{z})} dz + 0.5 \int_{-\infty}^{+\infty} q(\boldsymbol{z}) \cdot \log \frac{2q(\boldsymbol{z})}{p(\boldsymbol{z}) + q(\boldsymbol{z})} dz \quad (13)$$

Subsequently, we denote expectations and variances of $\mathcal{P}_{Z|0}$ and $\mathcal{P}_{Z|1}$ as $(\mu_0, \sigma_0)$ and $(\mu_1, \sigma_1)$, respectively. With the assumption that samples of each domain are symmetrically distributed about the centroid, we have $p_m = \max\{p(\boldsymbol{z})\} = p(\mu_0)$ and $q_m = \max\{q(\boldsymbol{z})\} = q(\mu_1)$. Thus if we use two variables $f$ and $g$ to denote PDF-s of $\mathcal{P}_{Z|0}$ and $\mathcal{P}_{Z|1}$, i.e., $f = p(\boldsymbol{z})$ and $g = q(\boldsymbol{z})$, in this case, we have $f \in (0, p_m]$ and $g \in (0, q_m]$. Based on the above analysis and notations, we can split Eq. (13) into 4 terms as follows,

$$
\begin{aligned}
I(\boldsymbol{z}; \boldsymbol{n}) = &\ 0.5 \int_0^{p_m} f \cdot \log \frac{2f}{f + g} df + 0.5 \int_0^{p_m} f \cdot \log \frac{2f}{f + g'} df \\
&+ 0.5 \int_0^{q_m} g \cdot \log \frac{2g}{f + g} dg + 0.5 \int_0^{q_m} g \cdot \log \frac{2g}{f' + g} dg
\end{aligned}
\quad (14)
$$

here the superscript $\prime$ indicates the right side of $f$ and $g$. Next, let us consider the Gaussian Kernel estimator $\text{MMD}(\mathcal{P}_{Z|0}, \mathcal{P}_{Z|1}; exp)$. The estimator consists of 3 terms, and we can easily conclude that $e^{-\|\boldsymbol{z}-\boldsymbol{z}'\|^2}$ decreases with the increase of $\|\boldsymbol{z} - \boldsymbol{z}'\|^2$. Note that 'MMD$(\mathcal{P}_{Z|0}, \mathcal{P}_{Z|1}; exp)$ increases to saturation' means at least one term increases while the other two terms remain unchanged or increased. For next proof, we need to mention Theorem 2 in (Sriperumbudur et al., 2009).

**Theorem 2** (Sriperumbudur et al., 2009). *Suppose $\{(X_i, Y_i)\}_{i=1}^N, X_i \in M, Y_i \in \{-1, +1\}, \forall i$ is a training sample drawn i.i.d. from $\mu$. Assuming the training sample is separable, let $f_{svm}$ be the solution to the program, $\inf\{\|f\|_{\mathcal{H}} : Y_i f(X_i) \geq 1, \forall i\}$, where $\mathcal{H}$ is an RKHS with measurable and bounded kernel $k$. If $k$ is characteristic, then*

$$\frac{1}{\|f_{svm}\|_{\mathcal{H}}} \leq \frac{\gamma_k(\mathbb{P}, \mathbb{Q})}{2} \quad (15)$$

where $\mathbb{P} := \frac{1}{d} \sum_{Y_i = +1} \delta_{X_i}$, $\mathbb{Q} := \frac{1}{d} \sum_{Y_i = -1} \delta_{X_i}$, $d$ is the sample quantity and $\delta$ represents the Dirac measure. This theorem provides a bound on the margin of hard-margin SVM in terms of MMD. Eq. (15) shows that a smaller MMD between $\mathbb{P}$ and $\mathbb{Q}$ enforces a smaller margin (i.e., a less smooth classifier, $f_{svm}$, where smoothness is measured as $\|f_{svm}\|_{\mathcal{H}}$). Besides, the Gaussian kernel is a measurable and bounded kernel function in an RKHS. According to this theorem and the nature of hard-margin SVM, we can easily obtained that variances $\sigma_0, \sigma_1$ of $\mathcal{P}_{Z|0}$ and $\mathcal{P}_{Z|1}$ are decreasing

and the difference between expectations $\mu_0$ and $\mu_1$ is increasing with the saturated increase of MMD, and this conclusion can be found in (Jegelka et al., 2009).

In the following, we will prove that $I(\boldsymbol{z};\boldsymbol{n})$ will increase when the difference between $\mu_0$ and $\mu_1$ increases. Due to the symmetry of PDFs, both $f$ and $g$ increase at the left side of their own expectation and decrease at the right side. Without the loss of generality, we assume that $\mu_0$ locates at the left side of $\mu_1$. For the first term of Eq. (14) which corresponds to the left interval of $f$, the value of $g$ is smaller than that of the case before increasing the difference between $\mu_0$ and $\mu_1$. Thus this term will increase when the difference between $\mu_0$ and $\mu_1$ increase. As for the right interval of $f$, the maximum value of $f + g'$ (in the neighborhood of $\mu_1$) comes later than that of case before increasing the difference, besides, the maximum value is also smaller. There, for the second term of Eq. (14), it will also increase with the increase of difference between $\mu_0$ and $\mu_1$. Similarly, the integration of $g$ follows the same trend with that of $f$. In this case, $I(\boldsymbol{z};\boldsymbol{n})$ will increase when the difference between $\mu_0$ and $\mu_1$ increases.

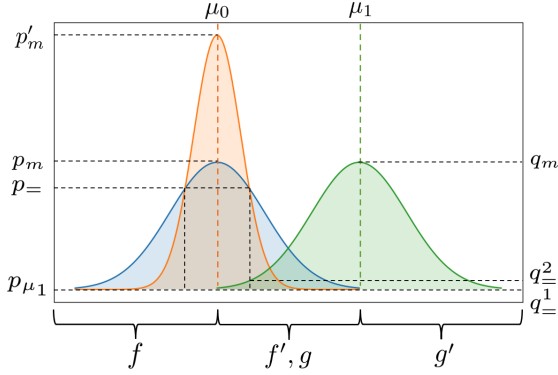
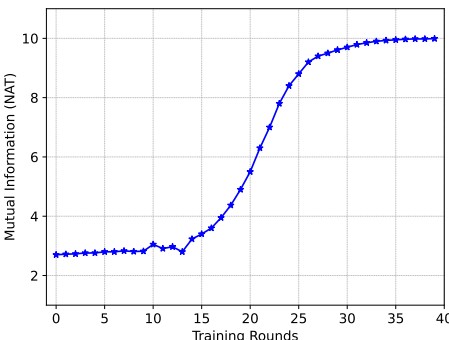

Figure 5: PDF-s of distribution $\mathcal{P}_{Z|0}$ and $\mathcal{P}_{Z|1}$. The blue curve is the PDF of $\mathcal{P}_{Z|0}$, and the green one is the PDF of $\mathcal{P}_{Z|1}$. The orange curve is the PDF of $\mathcal{P}_{Z|0}$ with a smaller variance (best view in color).

Figure 6: The change of mutual information $I(\boldsymbol{z};\boldsymbol{n})$ at every training round, and it is observed by the domain discriminator.

Next, we will prove that $I(\boldsymbol{z};\boldsymbol{n})$ will increase if the variance of either $\mathcal{P}_{Z|0}$ or $\mathcal{P}_{Z|1}$ decreases. Without the loss of generality, we assume the variance of $\mathcal{P}_{Z|0}$ decreases while the variance of $\mathcal{P}_{Z|1}$ remain unchanged. For the PDF of a distribution, if the variance decreases, the maximum value of PDF will increase, and there will be two points of intersection with the same value between the PDF-s. Such conclusions are easily proved since the integration of a PDF is always 1. We denote the new maximum value of $f$ as $p'_m$, and the value of two points of intersection as $p_=$. In addition, during the saturated increase of MMD, we can always find a pair of $\mu_0$ and $\mu_1$ that enables the left side of $g$ to intersect with the right side of $f$. With the notations in Figure 5. We can change Eq. (14) into

$$\text{Terms of } f = \int_0^{p_=} f \cdot \log \frac{2f}{f+g} df + \int_{p_=}^{p'_m} f \cdot \log \frac{2f}{f+g} df + \int_{p_m}^{p'_m} f \cdot \log \frac{2f}{f+g'} df$$
$$+ \int_{p_=}^{p_m} f \cdot \log \frac{2f}{f+g'} df + \int_{p_{\mu_1}}^{p_=} f \cdot \log \frac{2f}{f+g'} df + \int_0^{p_{\mu_1}} f \cdot \log \frac{2f}{f+g'} df \tag{16}$$

$$\text{Terms of } g = \int_0^{q_=^1} g \cdot \log \frac{2g}{f+g} dg + \int_{q_=^1}^{q_=^2} g \cdot \log \frac{2g}{f+g} dg$$
$$+ \int_{q_=^2}^{q_m} g \cdot \log \frac{2g}{f+g} dg + \int_0^{q_m} g \cdot \log \frac{2g}{f'+g} dg \tag{17}$$

According to the decrease of $\sigma_0$, we can conclude: the 1st, 4th, 6th terms of Eq. (16) and the 2nd term of Eq. (17) decrease, while the rest terms of Eq. (16) and Eq. (17) increase. For next proof, we denote a new function, $R(f,g) = f \cdot \log \frac{2f}{f+g}$, and we can get its first-order derivative is $\frac{\partial R}{\partial f} = \log \frac{2f}{f+g} + \frac{g}{f(f+g)}$. We can easily obtain that $\frac{\partial R}{\partial f} > 0$ when $f > g$. According to these

analysis, we can get that the decrease of the 1st term of Eq. (16) can be offset by the added increase of the 5th term of Eq. (16) and the 3rd term of Eq. (17); the decrease of the 4th term of Eq. (16) can be offset by the 2nd term of Eq. (16); the decrease of the 6th term of Eq. (16) can be offset by the 4th term of Eq. (17); the decrease of the 2nd term of Eq. (17) can be offset by the 3rd term of Eq. (16). Moreover, such offsets are overcompensation. In this case, we can prove that $I(z; n)$ will increase if the variance of either $\mathcal{P}_{Z|0}$ or $\mathcal{P}_{Z|1}$ decreases.

Considering the combination of the above two cases, if the difference between expectations increases and variances of two distributions decrease, the mutual information will increase. ∎

### A.2 Observe the Mutual Information

We follow the similar process of (Achille & Soatto, 2018) to observe the change of mutual information $I(z; n)$. To be specific, let $\Theta$ be a binary classifier that given representation $z$ and nuisance $n$ tries to predict whether $z$ is sampled from the distribution of one domain $\mathcal{P}_{Z|0}$ or another domain $\mathcal{P}_{Z|1}$. According to (Sønderby et al., 2016), if we train $\Theta$ with the loss of $\mathbb{E}_{z \sim \mathcal{P}_{Z|0}}(\log \Theta(z)) + \mathbb{E}_{z \sim \mathcal{P}_{Z|1}}(\log 1 - \Theta(z))$, there is always a Bayes-optimal $\Theta^*$,

$$\Theta^* = \frac{\mathcal{P}(z|n=0)}{\mathcal{P}(z|n=0) + \mathcal{P}(z|n=1)} \tag{18}$$

With Eq.(12), if we assume the $\Theta_0, \Theta_1$ trained with $\mathbb{E}_{z \sim \mathcal{P}_{Z|0}}(\log \Theta_0(z)) + \mathbb{E}_{z \sim \mathcal{P}_{Z|1}}(\log 1 - \Theta_0(z))$ and $\mathbb{E}_{z \sim \mathcal{P}_{Z|1}}(\log \Theta_1(z)) + \mathbb{E}_{z \sim \mathcal{P}_{Z|0}}(\log 1 - \Theta_1(z))$, respectively, are close to the optimal ones $\Theta_0^*, \Theta_1^*$, we have

$$I(z; n) = 0.5\mathbb{E}_{z \sim \mathcal{P}(z|n=0)} \log \frac{\mathcal{P}(z|n=0)}{\mathcal{P}(z)} + 0.5\mathbb{E}_{z \sim \mathcal{P}(z|n=1)} \log \frac{\mathcal{P}(z|n=1)}{\mathcal{P}(z)}$$
$$= 0.5\mathbb{E}_{z \sim \mathcal{P}(z|n=0)} \log 2\Theta_0(z) + 0.5\mathbb{E}_{z \sim \mathcal{P}(z|n=1)} \log 2\Theta_1(z) \tag{19}$$

With this approximation, we train $\Theta_0$ and $\Theta_1$ for the model of every NTL training round, and we get the curve of $I(z; n)$ shown in Figure 6 (MNIST). According to the figure, $I(z; n)$ is increasing during the overall training process, which is consistent with our intention.

## B Implementation Settings

### B.1 Network Architecture

To build the classification models, we use several popular architectures as the bottom feature extractor and attach them with fully-connected layers as the top classifier, which are shown in Table 5. Specifically, the backbone network of digits is VGG-11, that of CIFAR10 & STL10 is VGG-13, and we use both ResNet-50 and VGG-19 for VisDA. The classifiers of all models are the same, i.e., 3 linear layers with ReLU and dropout. As for the GAN in the augmentation framework, the generator $\mathcal{G}$ is made up of 4 ConvTranspose blocks and 2 Residual blocks, and the discriminator $\mathcal{D}$ consists of a feature extractor with 4 convolution layers, a binary classifier and a multi-class classifier. These two classifiers are composed of sequential fully-connected layers and share the same representations extracted from the front extractor. The detailed architecture is shown in Table 6 and 7.

Table 5: The architecture of classification models. 'img' is the dimension of representations extracted from the feature extractor.

| | |
|---|---|
| **Classifier** | Linear(256, K). 
 Linear(256, 256), ReLU, Dropout; 
 Linear(512*img*img, 256), ReLU, Dropout; |
| **Feature Extractor** | Backbone Network 
 (VGG-11/VGG-13/VGG-19/ResNet50) 
 [10:]. |
| **Feature Extractor** | Backbone Network 
 (VGG-11/VGG-13/VGG-19/ResNet50) 
 [0:10]. |

Table 6: The architecture of the generator $\mathcal{G}$. 'dim' is the dimension sum of the latent space and input label.

| Out | Tanh(). |
|---|---|
| **Conv4** | BatchNorm(3), ReLU. |
| | ConvTranspose2d(128, 3, 4, 2, 1); |
| **Conv3** | BatchNorm(128), ReLU. |
| | ConvTranspose2d(256, 128, 4, 2, 1); |
| **ResBlocks** | ResidualBlock(256) * 2. |
| **Conv2** | BatchNorm(256), ReLU. |
| | ConvTranspose2d(512, 256, 4, 2, 1); |
| **Conv1** | BatchNorm(512), ReLU. |
| | ConvTranspose2d(1024, 512, 4, 2, 1); |
| **Input** | Linear(dim, 1024). |

Table 7: The architecture of the discriminator $\mathcal{D}$.

| | Binary Classifier | Multiple Classifier |
|---|---|---|
| **Classifiers** | Linear(128, 1). | Linear(128, K). |
| | Linear(256, 128), ReLU, Dropout; | Linear(256, 128), ReLU, Dropout; |
| | Linear(512, 256), ReLU, Dropout; | Linear(512, 256), ReLU, Dropout; |
| **Feature Extractor** | Conv2d(256, 512, 3, 2, 1), LeakyReLU, Dropout. | |
| | Conv2d(128, 256, 3, 2, 1), LeakyReLU, Dropout; | |
| | Conv2d(64, 128, 3, 2, 1), LeakyReLU, Dropout; | |
| | Conv2d(3, 64, 3, 2, 1), LeakyReLU, Dropout; | |

### B.2 HYPER PARAMETERS

**Scaling factors and upper bounds.** As introduced in Section 3.1 of the main paper, there are two scaling factors $(\alpha, \alpha')$ that control the trade-off between the maximization of $I(\boldsymbol{z}; \boldsymbol{n})$ and the sufficiency property of the source domain. Here, we conduct experiments using different values ($\alpha = 0.01, 0.05, 0.10, 0.20, 0.50$ and $\alpha' = 0.01, 0.05, 0.10, 0.20, 0.50$), and evaluate their impact to the performance of NTL. For Target-Specified NTL, we select the combination of MNIST→USPS, STL10→CIFAR10 and VisDA-T→VisDA-V. For Source-Only NTL, we choose MNIST→Non-S, STL10→Non-S and VisDA-T→Non-S as the representatives to carry out experiments. The results are presented in Tables 8 and 9. It is easy to conclude that NTL can work effectively with different scaling factors. As for the upper bounds $(\beta, \beta')$, we set them for the sake of preventing the auxiliary domain loss and the MMD distance from dominating the optimization objective, affecting the convergence of training.

Table 8: The experiments on different values of the scaling factor $\alpha$.

| Scaling Factor $\alpha$ | Source/Target(%) | | | | |
|---|---|---|---|---|---|
| Cases/Values | 0.01 | 0.05 | 0.10 | 0.20 | 0.50 |
| MT/US | 98.6/14.3 | 98.2/14.2 | 97.9/14.5 | 97.7/14.2 | 97.6/14.3 |
| STL10/CIFAR10 | 88.0/11.1 | 87.6/12.2 | 85.4/09.4 | 82.9/08.7 | 83.0/12.1 |
| VisDA-T/VisDA-V | 93.8/08.8 | 94.9/08.9 | 94.4/11.3 | 93.5/08.7 | 93.2/09.8 |
| MT/Non-S | 98.9/19.0 | 98.8/13.4 | 98.9/14.7 | 98.2/13.4 | 98.5/13.3 |
| STL10/Non-S | 87.9/13.0 | 83.5/12.0 | 84.9/10.2 | 79.6/11.1 | 86.9/12.8 |
| VisDA-T/Non-S | 95.6/28.4 | 95.3/14.4 | 95.7/14.6 | 94.5/21.8 | 94.6/18.5 |

**Training parameters.** For the optimization of NTL, we utilize Adam as the optimizer, with learning $\gamma = 0.0001$ and batch size of 32. For all datasets, we randomly select 8,000 samples from their own training sets as the source data, and 1,000 samples from their own testing sets as the test data (if a dataset does not have test set, we select its test data from the training set without overlapping with the chosen 8,000 source samples). And the sample quantities of the source and auxiliary domain are always the same. In the training of adversarial augmentation, the optimizer is also Adam, and we set the learning rate to $\gamma = 0.0002$ with two decay momentums 0.5 and 0.999. The batch size is 64, and the dimension of the latent space fed to the generator is 256.

Table 9: The experiments on different values of the scaling factor $\alpha'$.

| Scaling Factor $\alpha'$ | Source/Target(%) | | | | |
|---|---|---|---|---|---|
| Cases/Values | 0.01 | 0.05 | 0.10 | 0.20 | 0.50 |
| MT/US | 99.3/14.2 | 98.8/14.3 | 97.9/14.5 | 99.2/14.2 | 99.3/14.2 |
| STL10/CIFAR10 | 87.2/08.8 | 82.0/10.3 | 85.4/09.4 | 85.8/10.8 | 85.4/11.4 |
| VisDA-T/VisDA-V | 96.6/11.0 | 95.3/09.0 | 94.4/11.3 | 92.5/08.9 | 96.4/10.3 |
| MT/Non-S | 98.9/09.4 | 98.7/13.5 | 98.9/14.7 | 99.1/14.3 | 99.3/15.3 |
| STL10/Non-S | 87.4/10.8 | 79.8/10.3 | 84.9/10.2 | 83.3/13.2 | 78.3/12.1 |
| VisDA-T/Non-S | 96.5/32.5 | 95.6/20.1 | 95.7/14.6 | 94.9/20.7 | 95.5/19.4 |

## B.3 Triggering and Authorization Patch

As mentioned in Section 4.1 and 4.2 of our main paper, we attach a patch on the data to utilize NTL for ownership verification and usage authorization. We create the patch in a simple way. Specifically, for the pixel of $i$-th row and $j$-th column in an RGB image, if either $i$ or $j$ is even, then a value of $v$ is added to the R channel of this pixel (the channel value cannot exceed 255). Intuitively, the patch is dependent on pixel values of each image. Thus the changes of feature space brought by the attachment of these patches for various images are not the same. In our experiments, if the content of the image is simple, e.g., MNIST, USPS and SVHN, the $v$ with a small value can shift the feature space sufficiently, but for more complicated images, we have to increase $v$ to enable source images attached with and without the patch differentiable. Specifically, we pick the value as follows: MNIST, USPS, SVHN ($v = 20$); MNIST-M, SYN-D, CIFAR10, STL10 ($v = 80$); VisDA ($v = 100$). As mentioned in the main paper, we will explore the unforgeability and uniqueness of patch generation in the further work.

## B.4 Implementation of Watermark Removal Approaches

In the Section 4.1 of the main paper, we implement 6 model watermark removal approaches to verify the effectiveness of NTL-based ownership verification. Here, we introduce how to implement these approaches. FTAL (Adi et al., 2018) is an approach that fine-tunes the entire watermarked model using the original training data. To implement it, we use 30% of training set that has been learned by NTL to fine-tune the entire model. When using RTAL (Adi et al., 2018), the top classifier is randomly initialized before fine-tuning. In our experiments, we load the feature extractor of the model trained with NTL and randomly initialize a classifier to attach on the extractor, and then use 30% of the training set to fine-tune this combined model. As for EWC (Chen et al., 2019), we use the code of (Chen et al., 2019) to compute the fisher information of network parameters and adjust the learning rate of fine-tuning. The data used by EWC is also 30% of the training set. Finally, AU (Chen et al., 2019) utilizes the watermarked model to pseudo label additional unlabeled samples from other similar domains, and these samples will be used to fine-tune the model together with the original training set. Following this principle, we use 30% of the training set and the same quantity of unlabeled samples from other domains (the proportion ratio between these two parties is 1:1) to fine-tune the model trained with our NTL. We conduct all fine-tuning methods for 200 epochs. For the watermark overwriting, we overwrites a new backdoor-based watermark (Zhang et al., 2018) on the model trained with NTL. Specifically, we attach a white corner ($3 \times 3$) as the backdoor trigger to $1/15$ of the training set, and follow the training approach of (Zhang et al., 2018) to write the watermark on the model. In addition, similar to other watermarking works (Rouhani et al., 2018), we also test if NTL-based verification is resistant to model pruning, and apply a layer-wise pruning method (Han et al., 2015) to prune 70% parameters of the model trained with NTL.

Table 10: The results (%) of authorizing model usage on CIFAR10 & STL10 and VisDA.

| Test | CIFAR10 | | STL10 | |
|---|---|---|---|---|
| Source | with Patch | without Patch | with Patch | without Patch |
| CIFAR10 | 85.9 ($\pm$0.97) | 11.5 ($\pm$1.29) | 42.3 ($\pm$1.57) | 13.5 ($\pm$2.01) |
| STL10 | 20.7 ($\pm$1.01) | 11.7 ($\pm$0.98) | 85.0 ($\pm$1.36) | 12.1 ($\pm$1.75) |
| Test | VisDA-T | | VisDA-V | |
| Source | with Patch | without Patch | with Patch | without Patch |
| VisDA-T | 93.2 ($\pm$1.12) | 20.9 ($\pm$1.04) | 22.5 ($\pm$1.04) | 17.3 ($\pm$1.98) |

Table 11: The experiment results (%) of VisDA on VGG-19.

| Test | VisDA-T | | VisDA-V | |
|---|---|---|---|---|
| Cases | with Patch | without Patch | with Patch | without Patch |
| Supervised Learning | - | 93.4 ($\pm$1.83) | - | 45.2 ($\pm$2.24) |
| Target-Specified NTL | - | 92.8 ($\pm$1.04) | - | 08.9 ($\pm$2.01) |
| Source-Only NTL | - | 92.4 ($\pm$1.17) | - | 11.3 ($\pm$1.00) |
| Ownership Verification | 10.1 ($\pm$0.90) | 93.0 ($\pm$0.98) | - | - |
| Model Authorization | 92.9 ($\pm$1.31) | 12.7 ($\pm$2.22) | 21.4 ($\pm$1.01) | 11.0 ($\pm$0.79) |

Table 12: The error range (%) of experiment results for Supervised Learning and Target-Specified NTL respectively (the left of '/' is Supervised Learning, and the right is Target-Specified NTL).

| Source/Target | MT | US | SN | MM | SD |
|---|---|---|---|---|---|
| MT | $\pm$0.37/$\pm$0.22 | $\pm$1.74/$\pm$1.61 | $\pm$1.89/$\pm$1.25 | $\pm$0.58/$\pm$0.96 | $\pm$1.97/$\pm$1.00 |
| US | $\pm$1.27/$\pm$1.73 | $\pm$0.10/$\pm$0.14 | $\pm$1.21/$\pm$1.00 | $\pm$1.64/$\pm$1.72 | $\pm$1.17/$\pm$1.23 |
| SN | $\pm$1.23/$\pm$1.00 | $\pm$1.67/$\pm$1.98 | $\pm$0.97/$\pm$0.94 | $\pm$0.86/$\pm$0.88 | $\pm$2.17/$\pm$1.47 |
| MM | $\pm$1.28/$\pm$0.79 | $\pm$1.78/$\pm$1.26 | $\pm$3.66/$\pm$1.08 | $\pm$1.78/$\pm$0.97 | $\pm$2.10/$\pm$1.90 |
| SD | $\pm$0.91/$\pm$1.79 | $\pm$1.61/$\pm$1.04 | $\pm$1.27/$\pm$1.07 | $\pm$1.19/$\pm$1.88 | $\pm$0.97/$\pm$0.61 |

## C  ADDITIONAL EXPERIMENTAL RESULTS

### C.1  AUGMENTATION DATA OF OTHER DATASETS

In the main paper, we present the augmentation data of MNIST, and in this section, we include the augmentation data of other datasets as follows: Figure 7 for USPS, Figure 8 for SVHN, Figure 9 for MNIST-M, Figure 10 for SYN-D, Figure 11 for CIFAR10, Figure 12, Figure 13 for VisDA-T.

### C.2  MODEL USAGE AUTHORIZATION ON CIFAR10 & STL10 AND VISDA

Here we present the experiment of authorizing the model usage on CIFAR10 & STL10 and VisDA, shown in Table 10. According to the results, the model performs well on the data attached with the authorized patch and has bad performance on all other samples.

### C.3  ADDITIONAL RESULTS OF VISDA ON VGG-19

To demonstrate the effectiveness of NTL on different network architectures, we also carry out experiments of VisDA on VGG-19. All other settings are the same as before, and the results are shown in Table 11. We can easily see that the performance is consistent with the aforementioned other experiments, which shows the wide applicability of NTL.

### C.4  ERROR BAR

We conduct all experiments with three random seeds (2021, 2022, 2023), and present the error range in this section. Table 12 is the error range of Target-Specified NTL corresponding to Table 1 of the main paper; Table 13 presents the error of experiments on Source-Only NTL corresponding to Table 3 of the main paper; Table 14 shows the error of model authorization which is presented as Table 4 in our main paper.

### C.5  THE IMPACT OF GAUSSIAN KERNEL BANDWIDTH

In our implementation, we utilize a series of Gaussian kernels to approximate MMD, which is implemented as the MK-MMD in Long et al. (2015). Specifically, the bandwidth in our used kernels is controlled by two parameters mul and num (we use mul = 2.0, num = 5 in our experiments presented in the main text of the paper). The bandwidth of these kernels is as follows:

$$\mathbf{B} = \left\{ \frac{\|x_1 - x_2\|^2 \cdot \text{mul}^{i - \lfloor \text{num}/2 \rfloor}}{(n^2 - n)} \right\}_{i=0}^{\text{num}-1} \tag{20}$$

where $x_1$ and $x_2$ are two input data batches with size $n$, and $\lfloor \cdot \rfloor$ extracts the integer part of the input. To investigate the impact of kernel bandwidth, we select a series of mul-s and num-s to conduct Source-Only NTL experiments on MNIST, CIFAR10 and VisDA-T, and the results are shown in Table 15. According to the results, we can observe that the performance difference between the source and target is nearly the same with different mul-s and num-s. As these two parameters

Table 13: The error range (%) of experiment results for Supervised Learning and Source-Only NTL respectively (the left of '/' is Supervised Learning, and the right is Source-Only NTL).

| Source/Non-S | MT | US | SN | MM | SD |
|---|---|---|---|---|---|
| MT | ±0.37/±0.13 | ±1.74/±1.23 | ±1.89/±1.22 | ±0.58/±0.21 | ±1.97/±1.04 |
| US | ±1.27/±1.09 | ±0.10/±0.34 | ±1.21/±0.77 | ±1.64/±0.78 | ±1.17/±0.79 |
| SN | ±1.23/±1.04 | ±1.67/±1.31 | ±0.97/±0.79 | ±0.86/±1.11 | ±2.17/±1.03 |
| MM | ±1.28/±0.31 | ±1.78/±1.67 | ±1.66/±1.75 | ±1.78/±0.54 | ±2.10/±1.12 |
| SD | ±0.91/±1.05 | ±1.61/±1.24 | ±1.27/±1.01 | ±1.19/±1.47 | ±0.97/±0.88 |

Table 14: The error range (%) of experiment results for authorizing usage of models trained with Source-Only NTL on digits.

| Source with Patch | Test | | | | | Test without Patch | | | | |
|---|---|---|---|---|---|---|---|---|---|---|
| | MT | US | SN | MM | SD | MT | US | SN | MM | SD |
| MT | ±0.43 | ±1.11 | ±1.89 | ±1.09 | ±1.37 | ±2.44 | ±1.22 | ±0.90 | ±1.74 | ±1.29 |
| US | ±1.73 | ±0.78 | ±0.88 | ±2.21 | ±2.60 | ±1.11 | ±0.73 | ±1.12 | ±1.71 | ±0.45 |
| SN | ±1.01 | ±1.04 | ±0.36 | ±0.97 | ±0.98 | ±1.17 | ±1.01 | ±0.92 | ±1.39 | ±0.51 |
| MM | ±1.29 | ±1.19 | ±0.34 | ±0.90 | ±1.20 | ±2.25 | ±0.87 | ±1.13 | ±1.23 | ±1.28 |
| SD | ±0.67 | ±1.29 | ±1.21 | ±1.82 | ±0.47 | ±1.39 | ±0.97 | ±1.38 | ±1.27 | ±2.61 |

directly determine the kernel bandwidth, these results demonstrate that the kernel bandwidth does not have a significant impact on NTL performance.

## D  POSSIBLE ATTACKS BASED ON NTL

Although we propose the Non-Transferable Learning for protecting the Intellectual Property in AIaaS, if the model owner is malicious, they can also utilize NTL to poison or implant backdoor triggers to their model evasively and release the model to the public. In the setting of applying Target-Specified NTL to verify the model ownership, the patch we used can also be regarded as a trigger for certain misclassification backdoor. From the results of ownership verification in the main paper, we can see the possibility of launching NTL-based target backdoor attacks. As for the case of Source-Only NTL, our objective is shaped like an universal poison attack by restricting the generalization ability of models. The results in our main paper demonstrate the feasibility of this poison attack. In addition, recently, there are more domain adaptation (DA) works about adapting the domain-shared knowledge within the source model to the target one without the access to the source data (Liang et al., 2020; Ahmed et al., 2021; Kundu et al., 2020). However, if the source model is trained with Source-Only NTL, we believe that these DA works will be ineffective. In other words, our NTL can be regarded as a type of attack to these source-free DA works.

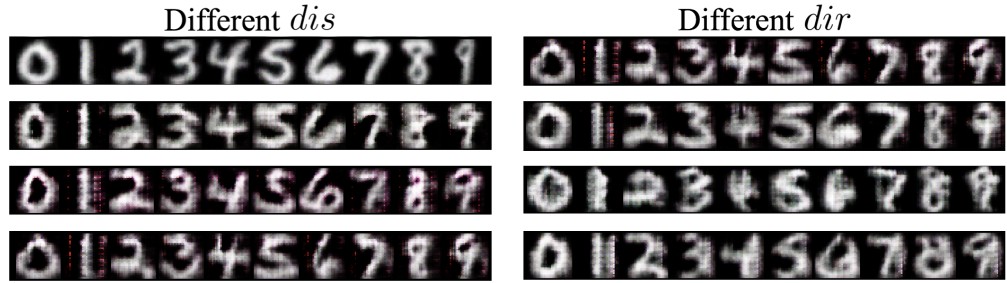

Figure 7: Augmentation results of USPS generated by the generative adversarial augmentation framework.

Table 15: The Source-Only NTL performance with different Gaussian kernel bandwidths (controlled by mul and num). The left of '/' is the performance on the source data, and the right is the average performance on the target data.

| Source/Bandwith | mul=1.0 | mul=2.0 | mul=3.0 | num=3 | num=5 | num=7 |
|---|---|---|---|---|---|---|
| | performance on the source / target data (%) | | | | | |
| MT | 98.8 / 14.5 | 98.9 / 14.7 | 98.7 / 14.6 | 98.8 / 14.6 | 98.9 / 14.7 | 98.8 / 14.7 |
| CIFAR10 | 87.7 / 10.3 | 87.8 / 10.2 | 87.8 / 10.3 | 87.7 / 10.5 | 87.8 / 10.2 | 87.7 / 10.4 |
| VisDA-T | 95.5 / 14.7 | 95.7 / 14.7 | 95.6 / 14.8 | 95.7 / 14.5 | 95.7 / 14.7 | 95.7 / 14.4 |

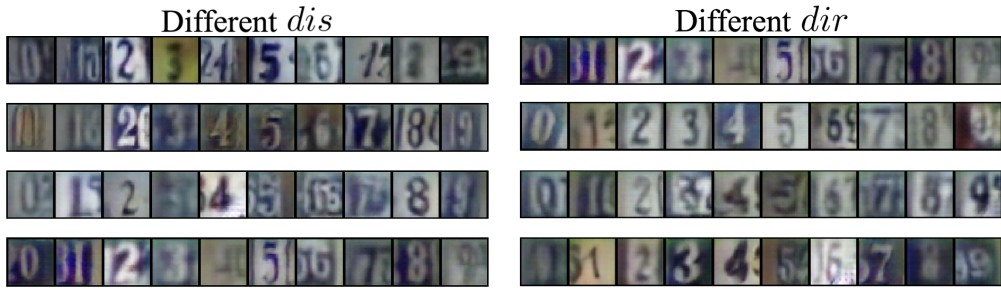

Figure 8: Augmentation of SVHN generated by the generative adversarial augmentation framework.

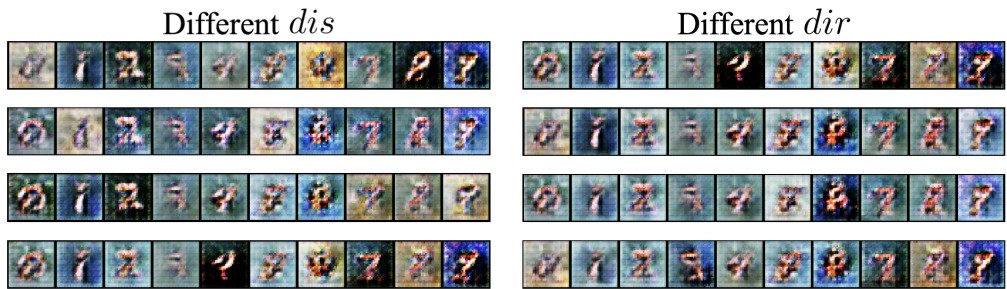

Figure 9: Augmentation results of MNIST-M generated by the generative adversarial augmentation framework.

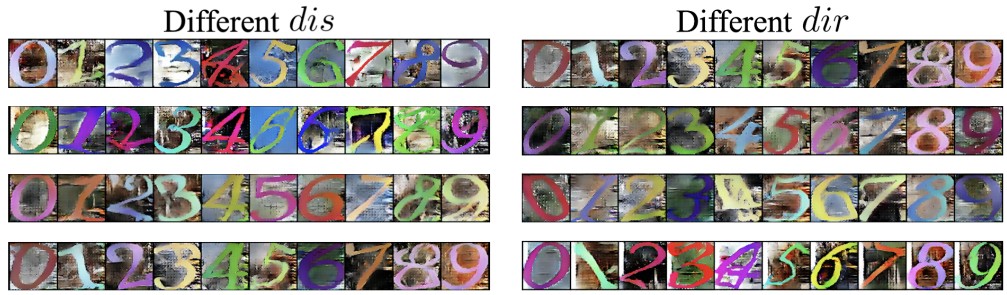

Figure 10: Augmentation results of SYN-D generated by the generative adversarial augmentation framework.

Different $dis$        Different $dir$

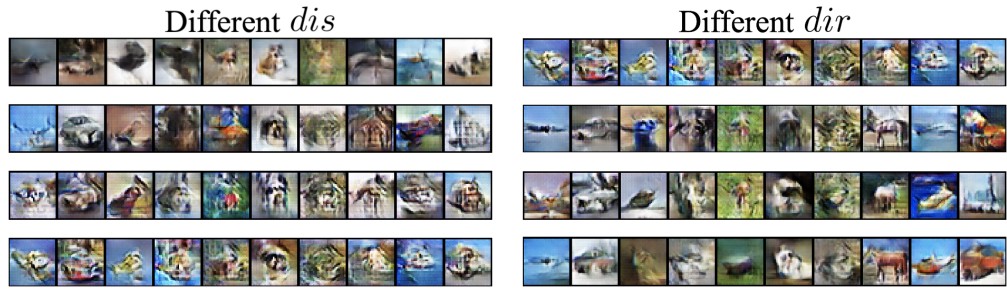

Figure 11: Augmentation results of CIFAR10 generated by the generative adversarial augmentation framework.

Different $dis$        Different $dir$

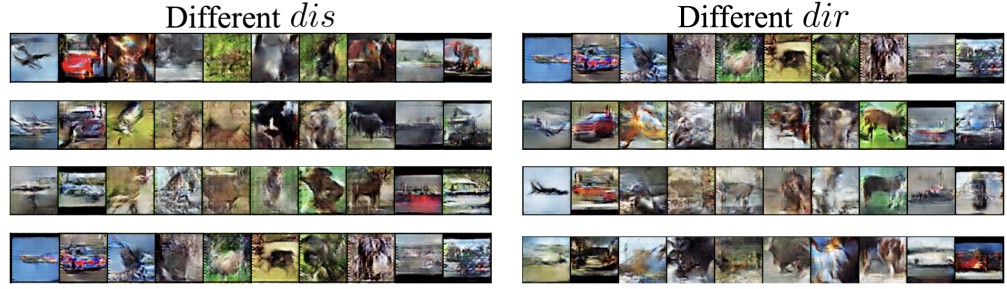

Figure 12: Augmentation results of STL10 generated by the generative adversarial augmentation framework.

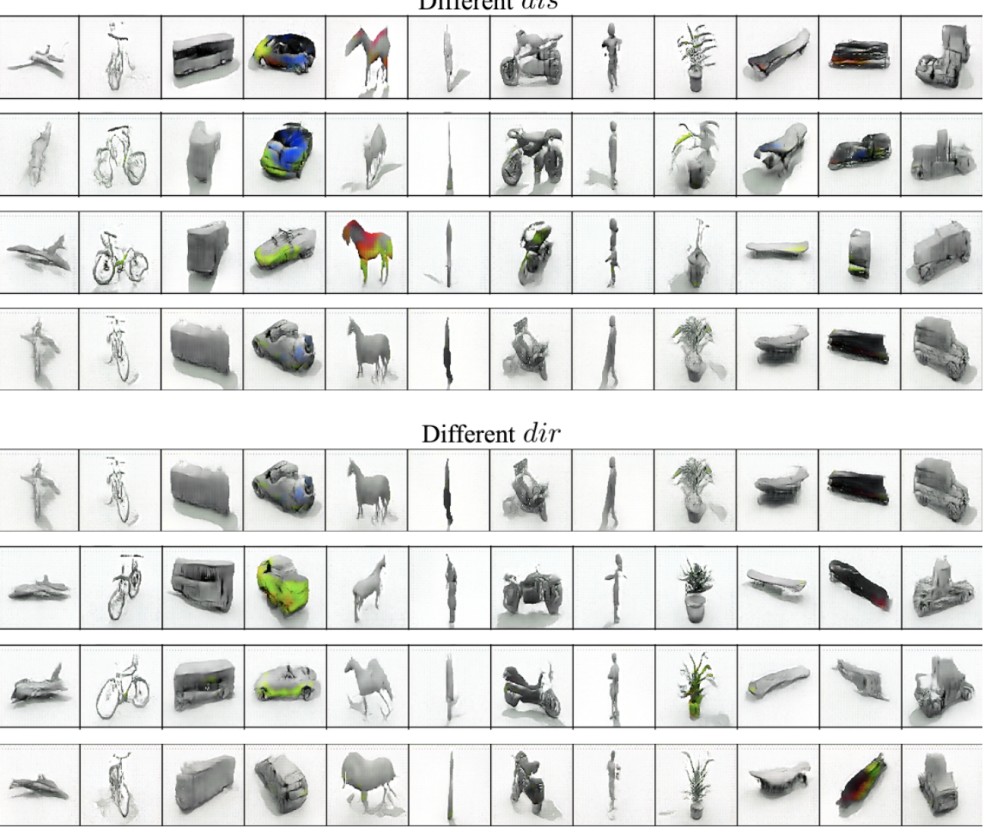

Figure 13: Augmentation results of VisDA-T generated by the generative adversarial augmentation framework.

