# OpenReview forum: "Non-Transferable Learning: A New Approach for Model Ownership Verification and Applicability Authorization"
_ICLR.cc/2022/Conference — ICLR 2022 Oral_

### Official Review · Reviewer_RSPY · 2021-10-26

**Correctness:** 4
**Technical Novelty And Significance:** 4
**Empirical Novelty And Significance:** 4
**Recommendation:** 8
**Confidence:** 5

**Main Review:**

This paper could be significantly improved via addressing the following issues:
1. In Table 1, what is the number of  training epoches when transfering MT to MM? Did you try to increase the epochs of fine-tuning? If you train for enough epochs, the model would eventually reach the original accuracy. The sensitivity analysis regarding the epoches of your fine-tuning is necessary when compared to training from scratch and the transfer learning from the original model to the target task.
2. The training complexity of using your NTL approach and the GAN training should be introduced in this paper? The computing time of the MMDs during each time step is at least twice your training time?
3. The propsoed methodology is well presented. However, the differences between the proposed model and realted SOTA works should be presented clearly.
4. Comparing Table 2 and Table 3, it can be seen that sometimes the source-only method shows greater performance compared to the target-specific method. The reasons why would this happen are interesting since providing the target-domain target should be more accurate when removing some part in the generalization space. However, the experiments seem does not agree with it.
5. A future research section should be added in the revision.

**Summary Of The Paper:**

Protecting the intellectual property of the trained models has received appealing attentions. Existing researches to protect  intellectual property fall into two major categories: ownership verification and usage authorization. To this end, the authors propose to utilize non-transferable learning to achieve both the goal of ownership verification and usage authorization. Extensive experiments on several representative datasets validate the effectiveness of the proposed method in terms of ownership verification.

Generally, this paper proposes a novel idea to address a practical problem in real-world applications, which could inspire many readers to follow it and have an important influence on the community of computer vision. I support the acceptance of this paper for a better ICLR conference.

**Summary Of The Review:**

This paper proposes an interesting question and gives the corresponding solution. I recommend the acceptance of this paper.

---

> ### Author Response · Authors · 2021-11-21
> **Response for Reviewer RSPY**
>
> >4. Comparing Table 2 and Table 3, it can be seen that sometimes the source-only method shows greater performance compared to the target-specified method. The reasons why would this happen are interesting since providing the target-domain target should be more accurate when removing some part in the generalization space. However, the experiments seem does not agree with it.
>
> Thanks for the comment. In our experiments of Target-Specified NTL, both the training and testing sets are subsets of the target domain, and they are not overlapping with each other. In this case, if there is a significant difference between the target training and testing sets, although the NTL training can effectively reduce the performance on the training set, the performance on the testing set may not decrease to the same extent. As for Source-Only NTL, our augmentation framework generates data with different directions and distances. It is possible that the generated data completely covers the target testing set, in which case the performance of Source-Only NTL can be better than that of Target-Specified NTL. There could also be other factors, e.g., various types of randomness. This is indeed an interesting aspect that we plan to investigate more in future work.
>
> >5. A future research section should be added to the revision.
>
> Thanks for the suggestion. We have included a brief discussion of future work in the revision (Section 5 is now Conclusion and Future Work), including suggested directions from the reviewers and some of our thoughts.

---

> > ### Comment · Reviewer_RSPY · 2021-11-29
> > **Thanks for your careful response.**
> >
> > The authors' response has addressed my concerns. I lean to accept this paper.

---

> ### Author Response · Authors · 2021-11-21
> **Response for Reviewer RSPY**
>
> >1. In Table 1, what is the number of training epochs when transferring MT to MM? Did you try to increase the epochs of fine-tuning? If you train for enough epochs, the model would eventually reach the original accuracy. The sensitivity analysis regarding the epochs of your fine-tuning is necessary when compared to training from scratch and transfer learning from the original model to the target task.
>
> Thank you very much for the question and comment. In experiments for Table 1, we set the number of training epochs as 50 for both Supervised Learning (left) and NTL (right). We tried increasing the number of training epochs to 100 on a number of different datasets and the results were similar. The only fine-tuning-related experiments in our work are fine-tuning-based watermark removal attacks (FTAL, RTAL, EWC, and AU), which are shown in Table 2, with the number of fine-tuning epochs set to 200. Indeed we should have a sensitivity analysis for the number of these fine-tuning epochs. Thanks for pointing this out. We have conducted additional experiments with different numbers of fine-tuning epochs for Table 2: 400, 600, 800, 1000 (in addition to the 200 epochs in the original submission). Taking the AU removal attack as an example, the CIFAR10 data with and without the trigger patch have the performance of 13.1/87.3 with 400 epochs, 13.0/87.1 with 600 epochs, 13.1/87.3 with 800 epochs, and 13.3/87.2 with 1000 epochs. We can observe a similar level of effectiveness for NTL under these different numbers of fine-tuning epochs. The results on MNIST and part of VisDA show a similar trend, and more experiments are ongoing. We plan to add a section in the Appendix once we obtain all the experimental results.
>
> >2. The training complexity of using your NTL approach and the GAN training should be introduced in this paper? The computing time of the MMDs during each time step is at least twice your training time?
>
> Thanks for the questions. NTL costs more time when compared with supervised learning (SL), and the additional overhead is mainly brought by MMD calculation and GAN training if it is the source-only case. In the experiments, we observe that the additional overhead is not very significant. For instance, in each training epoch of CIFAR10 on the VGG-13 backbone, NTL takes 57.2 seconds while SL takes 50.1 seconds. As for the training of our generative augmentation framework, the first stage (standard GAN training process) takes 40.8 seconds on average for each epoch. In the later augmentation stage, it takes 28.6 seconds on average for each epoch.
>
> >3. The proposed methodology is well presented. However, the differences between the proposed model and related SOTA works should be presented clearly.
>
> We are not completely sure which aspects the reviewer meant by related SOTA works. Below we provide some explanations based on our understanding of the context of the reviewer's comment and have made changes in the paper accordingly. If our understanding does not match what the reviewer meant, we are happy to further explain and improve it.
>
> For NTL methodology, to our best knowledge, we are the first to propose an optimization objective for reducing model generalization ability inspired by the Information Bottleneck Theory, and we are not aware of close SOTA works. At the beginning of the Methodology section, we have explained how NTL is inspired by the Information Bottleneck but addresses a different problem. As for the adversarial augmentation framework, although it is based on GAN, our aim is not to propose a new GAN but to design an effective augmentation method for NTL, and thus we did not discuss the SOTA works of GAN. We have added these explanations in Section 3 of the revision and made them bold, to clarify the difference between our work and the prior works, for both NTL and GAN. In addition, we have added more references in the Related Work section, on approaches for domain adaptation and works related to IP protection for DL models (in particular on Membership Inference Attack and Model Inversion Attack).

---

### Official Review · Reviewer_1wBe · 2021-10-31

**Correctness:** 4
**Technical Novelty And Significance:** 3
**Empirical Novelty And Significance:** Not applicable
**Recommendation:** 8
**Confidence:** 2

**Details Of Ethics Concerns:**

No particular concerns. The authors already addressed some in their submission.

**Main Review:**

Basically, the authors design a clever technique for learning nuisance-dependent representations. Such a representation can be made to perform accurately for a particular source domain, but poorly for another target domain. Furthermore, the authors design a GAN type technique for generating samples outside the source domain to serve as a kind of generic target domain. This is obviously important, as one cannot know to which target domain the model would be later adapted to.

This is a very interesting paper, although I have to say I'm not an expert in this topic at all. Most of the paper is really nicely written and is pretty easy to follow. The experimental verification is clear and detailed, but mostly limited to small images, so it's hard to say how it actually performs in some real-life scenarios.

Couple questions come to mind:
- Can you imagine uses of this to other kinds of models, e.g., language models, or is this mainly meaningful for image data?
- It sounds like an NTL representation by nature is highly vulnerable to training data privacy attacks, like membership inference. Have you considered if one could use the NTL representation to particularly efficiently generate samples from (something close to) the training data distribution?

**Summary Of The Paper:**

This paper introduces the idea of "non-transferable learning", which is roughly what the name indicates. The authors explain the value of this as a security/IP protection tool to protect the model from being used on unauthorized data. In addition, this presents a kind of attack against domain adaption works that try to improve generalization bounds without access to source data.

**Summary Of The Review:**

Non-transferable learning is an interesting idea to explore, and this is the first step in that direction. I can imagine that there will be a lot of follow-up ideas both for attacking this, as well as improving upon it. I would definitely recommend accepting this for ICLR.

---

> ### Author Response · Authors · 2021-11-21
> **Response for Reviewer 1wBe**
>
>
> >1. Can you imagine uses of this to other kinds of models, e.g., language models, or is this mainly meaningful for image data?
>
> Thank you very much for the question. Currently, our NTL mainly focuses on image-related tasks. Applying NTL to other models is indeed interesting and we have included it in future work discussion (Section 5) in the revision. In particular, it could be interesting and challenging to apply our NTL in language applications. The current version of NTL is not directly applicable as the GAN-based data augmentation method cannot generate NLP task data. Moreover, to our knowledge, generating language data of different domains could be difficult, and thus developing an NTL with restricted domain generalization ability in NLP tasks deserves an in-depth study. In addition to the domain NTL, we are very interested in restricting the language model to certain tasks within a particular language.
>
> It would also be interesting in future work to explore other image tasks such as semantic segmentation and object detection/tracking, where more complex datasets exist (in the current work for image classification, VisDA is the most complex dataset we can find in domain adaptation/generalization). Finally, another possible future direction is NTL for Multi-Task Learning, where we can restrict the language model to certain tasks. We have added a brief discussion of this in the revision as well.
>
> >2. It sounds like an NTL representation by nature is highly vulnerable to training data privacy attacks, like membership inference. Have you considered if one could use the NTL representation to particularly efficiently generate samples from (something close to) the training data distribution?
>
> Thanks for the question. We feel that this question inspires a very interesting direction to explore for future work, i.e., does the model with intellectual property protection have the same robustness to adversarial attacks as the model without such protection? Below are some of our initial thoughts and investigations.
>
> For membership inference attack (MI), to our knowledge, the attacking target is to determine whether a sample has been learned by the target machine learning model, where this sample is similar to the training data used in the learning process. We believe that what MI attack cares about are those samples that have correct predictions when they are forwarded to the target model. Taking a lung cancer diagnosis model as an example, the MI attack is interested in whether the data of a particular lung cancer patient has been used to train this diagnosis model, but not in whether a liver cancer patient has been learned by this model since the model intuitively outputs wrong predictions on liver cancer data. Therefore, considering our problem, we think that the MI attack could aim to determine whether a sample from the source domain has been learned by the target model. In our initial investigation, we forward the source domain data to both models trained with NTL and standard supervised learning and compare the cosine similarity between their predictions. We observe that the cosine similarity is very close to 1, which indicates that there is no significant prediction difference between models trained with NTL and standard supervised learning in terms of the source domain data. This could indicate that the model trained with our NTL is not more vulnerable to membership inference attacks (more quantitative investigations are needed in the future work).
>
> As for sample generation from the NTL representation, we assume that you mean a type of model inversion attack that aims to reconstruct the input data from its corresponding model prediction (please correct us if it is not the case). As aforementioned, the prediction of models trained with NTL and standard supervised learning on the same source domain data sample is very close, and therefore we think that the vulnerability of the model trained with NTL is similar to that of the model trained with standard supervised learning in terms of the source domain data. As for the data from other similar domains, intuitively, the model trained with NTL should not learn the semantic information within such data, and thus we hypothesize that there should not be significant information embedding in the model trained with NTL in terms of those similar domains. That is to say, NTL can likely protect the model from model inversion attacks to some extent, but more quantitative investigations are needed in the future work.

---

### Official Review · Reviewer_FcxC · 2021-11-01

**Correctness:** 4
**Technical Novelty And Significance:** 4
**Empirical Novelty And Significance:** 4
**Recommendation:** 8
**Confidence:** 4

**Main Review:**

Pros:
+ The research direction is promising and important in the real world. Nowadays, AI companies will train their own deep models with abundant labelled data that costs a lot of resources. Thus, it is a good timing to research how to protect these models, which have become very important and practical.
+ This paper proposed a method that can be effective solutions to both model verification and authorization, which is general and is promising to be applied in other applications.
+ This paper is easy to follow. Experiments are enough to support the claims made in this paper. A plus should be that experiments are conducted with 6 removal approaches over the digits, CIFAR10 & STL10, and VisDA datasets.

Cons:
- The presentation should be improved. The first paragraph in intro is too long. It is better to divided it into several paragraphs to better demonstrate the key points of this paper.
- I am not sure if it is necessary to list the contributions in the introduction. Such contributions have been described clearly in intro and abs. It seems that you do not need to restate them.
- Key related works are missing. For an AI company, they need to be aware of many adversarial attacks, such as reprogramming attacks, model-inversion attacks. These works are also related to IP protection of deep learning. It would be better to conclude these attacks as related works as well. Some discussions should be also added for general readers of ICLR.
- Some notations should be changed. For example, we will not use X or Y to present distributions, instead, we will use them to represent random variables. It is better to use \sP_X to represent the distribution corresponding to a random variable X. It is unnecessary to use GMMD, you can use MMD(P,Q; k), where k is a Gaussian kernel (you can follow the notations from recent deep kernel MMD papers).
- How many times do you repeat your experiments? I did not see error bar/STD values of your methods. This should be provided to verify that the experimental results are stable.
- If we consider to add bandwidth to your kernel function, how does the kernel bandwidth affect your results?

**Summary Of The Paper:**

In the era of deep learning, pre-trained models have been regarded as intellectual properties of AI companies. Thus, protecting these models has been more and more important. To achieve this aim, this paper proposes a non-transferable learning (NTL) method to capture the exclusive data representation in the learned model and restrict the model generalization ability to certain domains. This approach provides effective solutions to both model verification and authorization. Specifically:
1) For ownership verification, watermarking techniques are commonly used but are often vulnerable to sophisticated watermark removal methods. By comparison, the NTL-based ownership verification provides robust resistance to state-of-the-art watermark removal methods, as shown in extensive experiments with 6 removal approaches over the digits, CIFAR10 & STL10, and VisDA datasets.
2) For usage authorization, prior solutions focus on authorizing specific users to access the model, but authorized users can still apply the model to any data without restriction. The NTL-based authorization approach instead provides a data-centric protection, which is called applicability authorization, by significantly degrading the performance of the model on unauthorized data.
In general, this paper contributes a novel method to the field, and experiments verified the success of the proposed method.

**Summary Of The Review:**

In general, considering the significance of the researched problem, this paper can be accepted by the ICLR2022. However, some points should be clarified and strengthened in the revision. I would like to strongly support this paper if my concerns can be fully addressed.

---

> ### Author Response · Authors · 2021-11-21
> **Response for Reviewer FcxC**
>
> >1. The presentation should be improved. The first paragraph in the intro is too long. It is better to divide it into several paragraphs to better demonstrate the key points of this paper.
>
> Thank you very much for the suggestion. We have split the original first paragraph into two in the revision. The new first paragraph explains the importance of IP protection for deep learning models, and the second paragraph discusses existing methods for protecting the DL model IPs, explains why they are limited, and introduces our focus in this work. We also polished the language throughout these paragraphs and the rest of the section.
>
> >2. I am not sure if it is necessary to list the contributions in the introduction. Such contributions have been described clearly in intro and abs. It seems that you do not need to restate them.
>
> We have reorganized the text related to contribution and removed repetitive discussion in our revision. This actually gives us more space to discuss related and future work in the following sections. Thanks!
>
> >3. Key related works are missing. For an AI company, they need to be aware of many adversarial attacks, such as reprogramming attacks, model-inversion attacks. These works are also related to IP protection of deep learning. It would be better to conclude these attacks as related works as well. Some discussions should be also added for general readers of ICLR.
>
> We have included more discussions on relevant adversarial attacks in the related work section, including membership inference attack and model-inversion attack, and have made other minor changes throughout the section.
>
> >4. Some notations should be changed. For example, we will not use X or Y to present distributions, instead, we will use them to represent random variables. It is better to use \sP_X to represent the distribution corresponding to a random variable X. It is unnecessary to use GMMD, you can use MMD(P, Q; k), where k is a Gaussian kernel (you can follow the notations from recent deep kernel MMD papers).
>
> Thanks for pointing this out. We have modified the notations of distributions and MMD accordingly in the revision.
>
> >5. How many times do you repeat your experiments? I did not see the error bar/STD values of your methods. This should be provided to verify that the experimental results are stable.
>
> In the original submission, as mentioned in the second paragraph of Section 4 (right before Section 4.1), our experiments were repeated three times with different seeds (2021, 2022, 2023) and the average performance was reported in the main paper. We also included the error range (error bar) of our experiments in Appendix C.4. From the experimental results, we can observe that the performance of our method has only little fluctuation under different seeds, which shows the stability of our method.
>
> >6. If we consider adding bandwidth to your kernel function, how does the kernel bandwidth affect your results?
>
> Thanks for the question. In our implementation, we use a series of Gaussian kernels to compute the MMD approximator, which is implemented as the MK-MMD (specified in Appendix C.5). The bandwidth of the used kernels is determined at the high level by two parameters (mul and num). We select the source-only NTL cases of MNIST, CIFAR10, and VisDA-T as the representatives to conduct experiments with different kernel bandwidth (mul = 1.0, 2.0, 3.0 with num = 5, and num = 3, 5, 7 with mul = 2.0), and obtain the following results:
>
> MNIST (source / target): mul=1.0 (98.8 / 14.5); mul=2.0 (98.9/14.7); mul=3.0 (98.7 / 14.6)
> 			    num=3 (98.8 / 14.6); num=5 (98.9/14.7); num=7 (98.8 / 14.7)
>
> CIFAR10 (source / target): mul=1.0 (87.7 / 10.3); mul=2.0 (87.8 / 10.2); mul=3.0 (87.8 / 10.3)
> 			    num=3 (87.7 / 10.5); num=5 (87.8 / 10.2); num=7 (87.7 / 10.4)
>
> VisDA-T (source / target): mul=1.0 (95.5 / 14.7); mul=2.0 (95.7 / 14.7); mul=3.0 (95.6 / 14.8)
> 			    num=3 (95.7 / 14.5); num=5 (95.7 / 14.7); num=7 (95.7 / 14.4)
>
> From the above results, we can see that the kernel bandwidth does not seem to have a significant impact on NTL performance. We have added the above results in the revision as Appendix C.5.

---

> > ### Comment · Reviewer_FcxC · 2021-11-25
> > **Thank you for the clarification**
> >
> > I have carefully read the responses and other reviewers' comments. My concerns have been properly addressed. I think this paper will contribute a lot to the machine learning field. Thus, I decide to raise my score to 8.

---

### Author Response · Authors · 2021-11-21
**Thanks for All Reviewers**

We would like to thank all the reviewers for their insightful comments and constructive suggestions. Below we provide our detailed response to each reviewer. We have also uploaded a revised version of our submission, with major changes highlighted in blue (there are also other minor changes on wording and typos). We look forward to further feedback and discussion.

---

### Public Comment · ~Junfeng_Guo2 · 2022-03-01
**Verification of Eq.(2)**

Hi, thanks for your interesting work! According to your released code, I think Eq.(2) in current version should be **min** not **max**.

---

### Decision · Program_Chairs · 2022-01-20

**Decision:**

Accept (Oral)

**Comment:**

The paper addresses two important aspects of deep learning: model transferability and authorization for use. It presents original solutions for both of these problems. All of the reviewers agree that the paper is a valuable contributions. Minor concerns and critical remarks have been addressed by the authors during the discussion.